# Controlling the dimension of the quantum resonance in CdTe quantum dot superlattices fabricated via layer-by-layer assembly

TaeGi Lee[1], Kazushi Enomoto [2], Kazuma Ohshiro[1], Daishi Inoue[2], Tomoka Kikitsu[2], Kim Hyeon-Deuk[3], Yong-Jin Pu [2✉] & DaeGwi Kim [1✉]

In quantum dot superlattices, wherein quantum dots are periodically arranged, electronic states between adjacent quantum dots are coupled by quantum resonance, which arises from the short-range electronic coupling of wave functions, and thus the formation of minibands is expected. Quantum dot superlattices have the potential to be key materials for new optoelectronic devices, such as highly efficient solar cells and photodetectors. Herein, we report the fabrication of CdTe quantum dot superlattices via the layer-by-layer assembly of positively charged polyelectrolytes and negatively charged CdTe quantum dots. We can thus control the dimension of the quantum resonance by independently changing the distances between quantum dots in the stacking (out-of-plane) and in-plane directions. Furthermore, we experimentally verify the miniband formation by measuring the excitation energy dependence of the photoluminescence spectra and detection energy dependence of the photoluminescence excitation spectra.

[1] Department of Applied Physics, Osaka City University, Osaka 558-8585, Japan. [2] RIKEN Center for Emergent Matter Science (CEMS), Saitama 351-0198, Japan. [3] Department of Chemistry, Kyoto University, Kyoto 606-8502, Japan. ✉email: yongjin.pu@riken.jp; tegi@a-phys.eng.osaka-cu.ac.jp

Semiconductor quantum dots (QDs) have high photo-luminescence (PL) quantum yields (QYs) at room temperature, and their absorption and PL energies can be tuned by changing their sizes[1]. Therefore, they have potential applications in displays[2,3], bioimaging[4,5], and optoelectronic devices such as photodetectors[6] and solar cells[7,8]. Novel optical properties based on the interaction between QDs are realised when they are sufficiently close to each other[1,9–12]. In particular, only when the surface-to-surface distance between QDs becomes shorter than ~2 nm, the quantum resonance, which arises from the short-range electronic coupling of wave functions, occurs between adjacent QDs[11,12]. When such quantum resonance appears in QD superlattices (QDSLs) in which colloidal QDs are regularly arranged, the formation of coupled electronic states (minibands) is expected. These minibands greatly improve charge transport properties; therefore, the fabrication of new optoelectronic devices utilising inter-QD interactions, such as field-effect transistors and highly efficient solar cells, have attracted considerable attention in recent years[13–19].

Murray et al. were the first to report the fabrication of QDSLs, and they produced CdSe QDSLs by tailoring the composition of the dispersing medium to slowly destabilise the colloidal QD dispersion as the solvent evaporated[1]. Following this work, researchers have reported various methods for fabricating QDSLs such as drop-casting colloidal QD solution onto quartz substrates[20], ligand exchange[21], and cluster-assisted assembly[22]. These articles mainly discussed the fabrication process of the QDSLs.

The formation of minibands in QDSLs requires quantum resonance between adjacent QDs. However, to synthesise colloidal QDs using the hot injection method, long-chain molecules such as trioctylphosphine oxide (TOPO) or oleic acid (OA) have been usually used as ligands[23,24], while the long ligands prevent short-range quantum resonance[25]. To solve this problem, the long ligands are usually exchanged with short-chain ligands such as ethanedithiol (EDT) or propanedithiol (PDT)[17]. Recently, exchange with exciton delocalised ligands that lower the QDSL potential barrier has been proposed[26–29]. However, the ligand exchange process degrades the PL properties of QDs[30]; thus, the quantum resonance must be realised between adjacent QDs in QDSLs without such ligand exchange.

One promising method is the use of water-soluble QDs because short-chain ligands such as thioglycolic acid (TGA), 3-mercaptopropionic acid (MPA), and N-acetyl-L-cysteine (NAC) are often used to synthesise water-soluble QDs[11,31–36]. These ligands are 0.4–0.6 nm long and are shorter than the TOPO and OA used to synthesise oil-soluble QDs. Thus, water-soluble QDs offer the advantage of reducing the distance between QDs[37]. In fact, we previously found the quantum resonance in the three-dimensional (3D) QDSLs composed of NAC-capped CdTe QDs deposited by the layer-by-layer (LBL) method[11]. The LBL assembly process is a simple and useful method for preparing highly homogeneous multilayer structures of water-soluble QDs[11,38–40]. This method is advantageous because the inter-layer space between homogeneous QD monolayers can be precisely controlled within a single nanometre accuracy using a spacer layer composed of oppositely charged polyelectrolytes deposited during the LBL assembly[11,40]. The quantum resonance has been clearly observed between the CdTe QD layers in the stacking direction without ligand exchange because the QDs were synthesised using a short NAC ligand from the beginning, which enables the sufficient shortening of the distance between QDs. Therefore, the intrinsic optical properties can be investigated based on the quantum resonance between adjacent QDs without the ligand exchange. The formation of miniband in CdSe[13–15], InAs[16], and PbSe[17,18] QD solids has been discussed mainly from

the viewpoint of transport properties. In contrast, the formation of minibands derived from the quantum resonance between QDs by our approach can be discussed from the viewpoint of optical properties such as absorption and PL properties.

In the CdTe QDSLs deposited via the LBL method, the inter-QD surfaces become shorter than 1 nm in the in-plane direction[11]; thus, the quantum resonance is considered to occur not only in the stacking direction but also in the in-plane direction. In other words, the 3D quantum resonance appears in the 3D QDSLs composed of CdTe QDs. As mentioned above, during the LBL assembly, the QD interlayer distance can be controlled by the thickness of the spacer layer. If this thickness is 2 nm or more, only the in-plane (i.e. two-dimensional (2D)) quantum resonance occurs, as in the QD monolayer structure (Fig. 1). Furthermore, if the in-plane QD distance is increased, that is, if the in-plane QD density ($\sigma_{in\text{-}plane}$) becomes low, the quantum resonance in the in-plane direction is suppressed. When the QD multilayer structures are fabricated under these conditions, the quantum resonance may be realised only in the stacking direction, that is, one-dimensional (1D) quantum resonance (Fig. 1), which is connected to the motivation of this study. In this paper, we report the successful control of the quantum resonance dimensions. In addition, we evaluate the formation of minibands in the QDSLs, in which the 1D, 2D, and 3D quantum resonance takes place, by measuring both the excitation energy dependence of the PL spectra and detection energy dependence of the PL excitation (PLE) spectra.

## Results

**2D quantum resonance in QD monolayers**. We used the NAC-capped CdTe QDs with a diameter $d = 3.4 \pm 0.3$ nm. The typical transmission electron microscopy (TEM) image and size histogram are shown in Supplementary Fig. 1. Figure 2a shows the absorption spectra of various CdTe QD monolayers prepared using colloidal QD solutions with different concentrations. The absorption spectra were obtained by correcting the baseline in the measured extinction spectra. The details of the analysis of the absorption spectra are described in Supplementary Information (Supplementary Figs. 2 and 3). We define $OD_{sol}$ as the optical density (OD) of the first absorption peak in the colloidal solution, measured using a cell with an optical path length of 10 mm. We plotted the OD of the absorption peak of the monolayers ($OD_{mono}$) against $OD_{sol}$, as shown in Fig. 2b. In the low-concentration region (i.e. $OD_{sol} < 0.15$), $OD_{mono}$ increases linearly. This finding indicates that $\sigma_{in\text{-}plane}$ of the monolayers can be controlled by changing the concentration of the colloidal solution. In addition, in the high-concentration region ($OD_{sol} > 0.15$), $OD_{mono}$ is saturated, which indicates that the QDs are closely packed in the monolayer without any space for accepting another QD.

We focused on the absorption peak energy of the monolayers. Figure 2c shows the dependence of the absorption peak energy of the monolayers on $OD_{mono}$. When $OD_{mono}$ is 0.005 or less, the absorption peak energy of the monolayer is nearly constant and coincides with the absorption peak energy of the colloidal solution in which the QDs are randomly dispersed. This observation indicates that QDs in the monolayers with low $\sigma_{in\text{-}plane}$ do not couple with each other, and the absorption properties of isolated individual QDs are observed. On the other hand, when $OD_{mono}$ is 0.005 or more, the absorption peak shifts to a lower energy as $\sigma_{in\text{-}plane}$ increases. This low-energy shift can be attributed to the in-plane quantum resonance between adjacent QDs.

We recorded the in-plane X-ray diffraction (XRD) patterns of the closely packed monolayer to confirm the periodicity in the in-

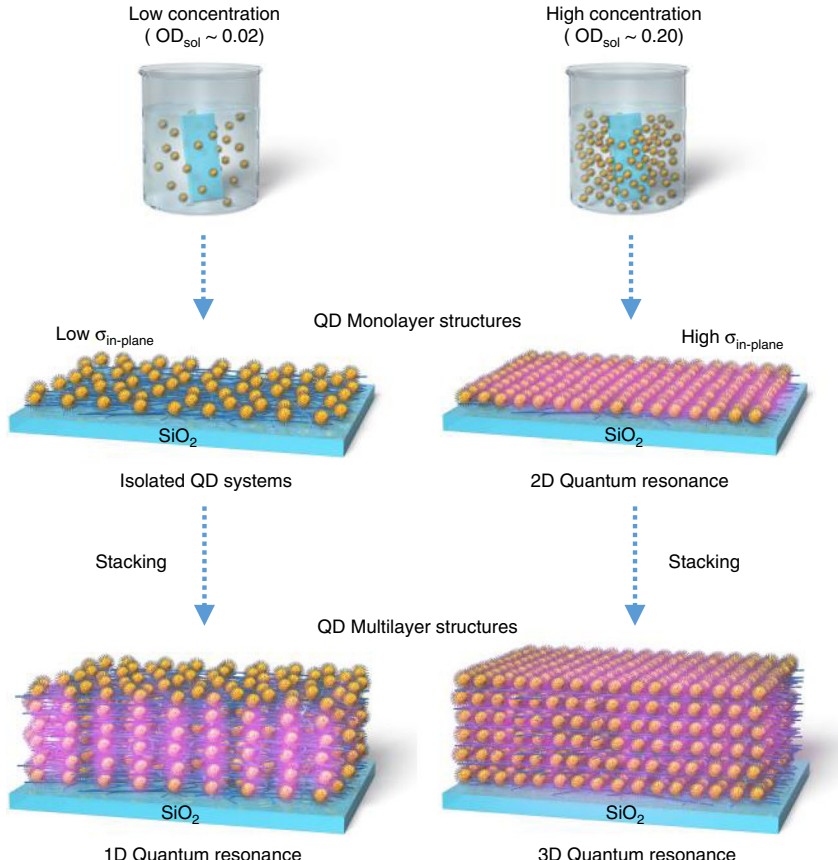

**Fig. 1 Schematics of quantum dot (QD) layer structures in which the 1D, 2D, and 3D quantum resonances occur.** We controlled the in-plane QD density ($\sigma_{\text{in-plane}}$) by changing the concentration of the QD solution and hence the optical density of the QD solution ($OD_{\text{sol}}$) using a layer-by-layer assembly method. The left and right sides of the figure represent QD layer structures with low and high $\sigma_{\text{in-plane}}$, respectively. If $\sigma_{\text{in-plane}}$ is low, the quantum resonance in the in-plane direction is suppressed (isolated QD systems). When QD multilayer structures are fabricated under the conditions of low $\sigma_{\text{in-plane}}$, the quantum resonance occurs only in the stacking direction, that is, the 1D quantum resonance is realised. In the QD monolayer structure with high $\sigma_{\text{in-plane}}$, only the in-plane 2D quantum resonance occurs. In the QD multilayer structures with high $\sigma_{\text{in-plane}}$, the quantum resonance appears not only in the stacking direction but also in the in-plane direction. In other words, the 3D quantum resonance occurs in the 3D QD superlattices. The coupled electronic states are represented by translucent purple colour.

plane direction as shown in Fig. 2d. The diffraction peak appears at $2\theta$ angle of 2.3°, which means that the QD monolayer has an in-plane spatial period of ~3.9 nm, which is almost similar to the QD diameter. Therefore, the XRD results indicate that the QDs are closely packed and periodically arrayed in the plane; therefore, we successfully fabricated the 2D QDSLs. Because the average diameter of the CdTe QDs is 3.4 nm, the surface-to-surface distance between QDs is estimated to be ~0.5 nm. As discussed in ref. [11], the quantum resonance arises from the coupling of the wave functions between the adjacent QDs, which is a short-range interaction, and its strength weakens exponentially with the increase in surface-to-surface distance between QDs. When the QD interlayer distance is larger than 2 nm, the absorption peak hardly shifts. Therefore, that the quantum resonance can occur only when the QD distance is ~2 nm or less, and the distance of 0.5 nm is sufficiently short to realise miniband formation.

The in-plane quantum resonance can be concluded to cause the low-energy shift in the absorption peak energy of the 2D QDSL ($OD_{\text{sol}} > 0.005$) relative to the absorption peak energy of the QD monolayer in which the QDs are randomly dispersed in the plane ($OD_{\text{sol}} < 0.005$), as shown in Fig. 2c. The absorption energy decreased by 19 meV, which can be considered as the coupling energy caused by the in-plane quantum resonance. We define this energy shift as $\Delta E_{\text{in-plane}}$ (=19 meV), as schematically shown in Supplementary Fig. 4.

The quantum resonance between adjacent QDs greatly depends on the distance between the QDs. To control the in-plane quantum resonance, the distance between QDs in the QD monolayer must be carefully controlled within a single nanometre accuracy. We determined the average surface-to-surface distance between QDs by examining scanning transmission electron microscopy (STEM) images of the monolayers. Figure 3a, b show the STEM images of the monolayers fabricated under the conditions without ($OD_{\text{sol}} = 0.02$) and with ($OD_{\text{sol}} = 0.15$).the occurrence of in-plane quantum resonance, respectively. In the images, we directly measured 120 centre-to-centre distances between QDs as shown in the inset in Fig. 3a; the corresponding histograms of the centre-to-centre distances are shown in Fig. 3c, d. From the histograms, the average centre-to-centre distances are estimated to be 5.6 nm with $OD_{\text{sol}} = 0.02$ and 3.9 nm with $OD_{\text{sol}} = 0.15$. The average diameter of the CdTe QDs is 3.4 nm; therefore, the average surface-to-surface distances can be estimated to be 2.2 nm with $OD_{\text{sol}} = 0.02$ and 0.5 nm with $OD_{\text{sol}} = 0.15$. In addition, the variance of the average surface-to-surface distances is larger with $OD_{\text{sol}} = 0.02$ than with $OD_{\text{sol}} = 0.15$, implying that QDs are dispersed not periodically but quite randomly in the former compared to the latter. The results in Figs. 2 and 3 clearly demonstrate that by merely changing $\sigma_{\text{in-plane}}$, we can effectively control the fabrication of either the

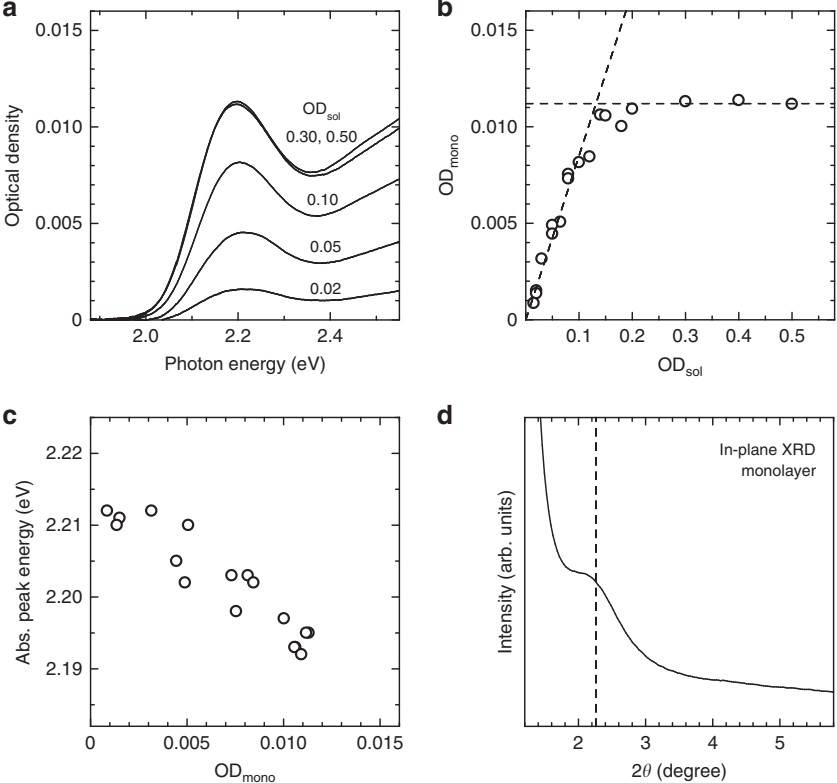

**Fig. 2 Absorption properties and X-ray diffraction (XRD) structural analyses of quantum dot monolayers. a** Absorption spectra of CdTe quantum dot monolayers fabricated from colloidal quantum dot solutions with different concentrations. **b** Optical densities (ODs) of the absorption peaks of the monolayers ($OD_{mono}$) versus the OD of the absorption peak of the colloidal quantum dot solution ($OD_{sol}$). One of the dashed lines is proportional to $OD_{sol}$, which shows that the $OD_{mono}$ is proportional to $OD_{sol}$. The other dashed line shows the value at which the $OD_{mono}$ is saturated. **c** Dependence of the absorption peak energy of the quantum dot monolayers on $OD_{mono}$. **d** In-plane XRD pattern of the CdTe quantum dot monolayer with high in-plane quantum dot density ($\sigma_{in\text{-}plane}$), which exhibits 2D quantum resonance. The dashed line shows the peak value.

isolated QD systems without any quantum resonance or the 2D QDSLs with in-plane quantum resonance.

**1D and 3D quantum resonance in QD multilayers**. Next, we discuss the absorption properties of the multilayer CdTe QD structures (hereafter referred to as 'QD multilayers' for simplicity) with low $\sigma_{in\text{-}plane}$ in which the in-plane quantum resonance does not occur and the QD multilayers with high $\sigma_{in\text{-}plane}$ in which the in-plane quantum resonance occurs. We prepared the QD multilayers with low and high $\sigma_{in\text{-}plane}$ by using the colloidal solutions of $OD_{sol} = 0.02$ and 0.20, respectively. Figure 4a, b show the corresponding absorption spectra of the QD multilayers prepared with these two different optical densities. Figure 4c shows the OD of the absorption peak in the QD multilayer ($OD_{multi}$) as a function of the number of layers ($n$). $OD_{multi}$ linearly increases with the number of layers, which suggests that QDs can be stacked at almost the same density in each layer. In addition, the difference in the increasing slope indicates that the $\sigma_{in\text{-}plane}$ values are different in the QD multilayers that are fabricated under the conditions of the different $OD_{sol}$ values. Figure 4d shows the absorption peak energy of the QD multilayers as a function of $n$. Notably, the absorption peak energy decreases as $n$ increases in the QD multilayers with high $\sigma_{in\text{-}plane}$. Also, in the QD multilayers with low $\sigma_{in\text{-}plane}$, the absorption peak shifts to the lower energy with an increase in $n$.

We discuss the amount of absorption energy shift. We define $\Delta E_{stacking}$ as the difference in the absorption peak energies between the QD monolayer with $n = 1$ and QD multilayer with $n$

$= 5$ in which the absorption energy stops shifting. In the samples with low and high $\sigma_{in\text{-}plane}$, the $\Delta E_{stacking}$ values are 22 and 20 meV, respectively, which are almost the same. In the QD multilayers with low $\sigma_{in\text{-}plane}$, although the shift of the absorption peak energy stops at $n = 5$, the peak energy does not approach the same number of layers in the QD multilayers with high $\sigma_{in\text{-}plane}$. This result indicates that the red shift in the QD multilayers prepared with low $\sigma_{in\text{-}plane}$ is not due to increase in $\sigma_{in\text{-}plane}$. The QD multilayers were prepared by the LBL assembly of the cationic polymer and CdTe QDs. It is considered that the presence of the polymer layer makes it possible to fabricate the QD multilayers with low $\sigma_{in\text{-}plane}$ without increasing $\sigma_{in\text{-}plane}$ even at $n = 3$–5. These results strongly support that the quantum resonance occurs in the stacking direction not only in the QD multilayers with high $\sigma_{in\text{-}plane}$ but also in the QD multilayers with low $\sigma_{in\text{-}plane}$. Therefore, 1D quantum resonance occurs in the stacking direction in QD multilayers prepared with low $\sigma_{in\text{-}plane}$.

We also measured out-of-plane XRD patterns to confirm the periodic structural ordering in the stacking direction. Figure 5 shows the out-of-plane XRD patterns of the CdTe QD multilayers (30 layers) prepared under with high and low $\sigma_{in\text{-}plane}$. Notably, the diffraction peak is observed at almost the same $2\theta$ angle of 2.5° in the both samples prepared under these two conditions. These results indicate that both of the CdTe QD multilayers exhibit the same structural regularity with a period of ~3.5 nm in the stacking direction. The coincidence of the diffraction angles suggests that they are aligned with the same spatial period in the stacking direction regardless of $\sigma_{in\text{-}plane}$. In other words, QDs are

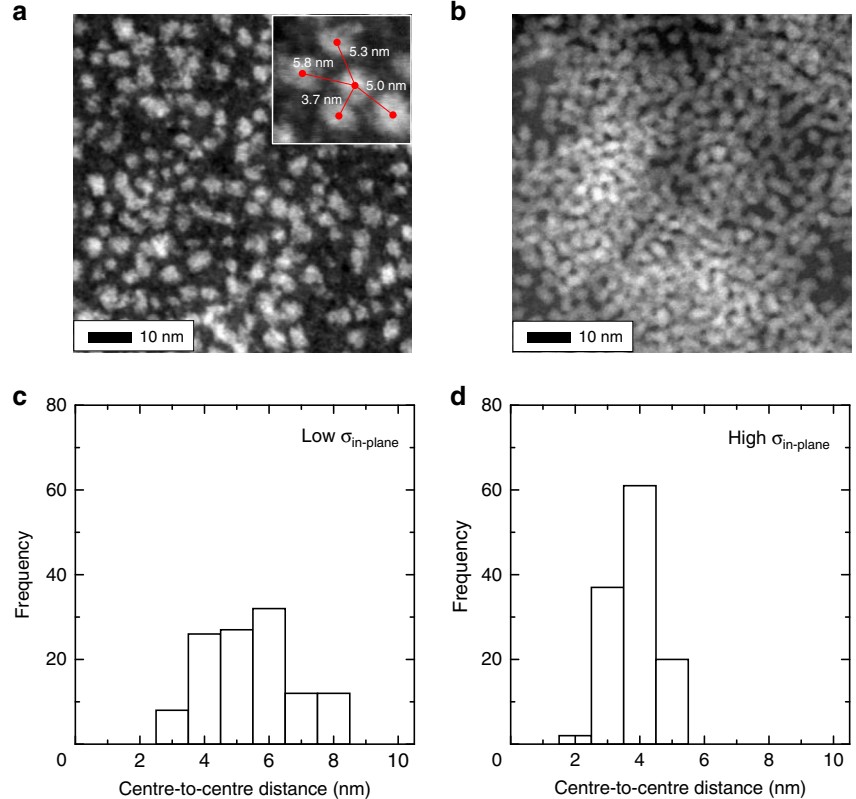

**Fig. 3 Scanning transmission electron microscopy images of quantum dot monolayers. a, b** Scanning transmission electron microscopy images of quantum dot monolayers with low **a** and high **b** in-plane quantum dot density ($\sigma_{\text{in-plane}}$), which were fabricated with $OD_{\text{sol}} = 0.02$ and 0.15, respectively. The inset in the figure is the enlarged image. The red dots point to the centre of the quantum dots, and the red lines connect the points. The numbers shown in white indicate the length of the red lines, which are centre-to-centre distance between quantum dots. **c, d** Histograms of the centre-to-centre distances between quantum dots in the quantum dot monolayers with low **c** and high **d** $\sigma_{\text{in-plane}}$.

regularly arranged in the stacking direction even in QD multilayers with low $\sigma_{\text{in-plane}}$. Actually, as shown in Fig. 4d, the 1D quantum resonance occurs in the stacking direction even in the QD multilayers with low $\sigma_{\text{in-plane}}$. In the CdTe QD multilayers prepared with high $\sigma_{\text{in-plane}}$, the quantum resonance occurs in both the in-plane and stacking directions, that is, the 3D quantum resonance. In the CdTe QD monolayer prepared with high $\sigma_{\text{in-plane}}$, the 2D quantum resonance occurs only in the in-plane direction. Therefore, we can control the dimension of the quantum resonance by changing $\sigma_{\text{in-plane}}$ and the number of layers in the LBL assembly.

Supplementary Fig. 4 schematically shows the energy shift of the absorption peak due to the quantum resonance. It was revealed that $\Delta E_{\text{in-plane}}$, which is the energy shift due to the in-plane quantum resonance, is 19 meV as discussed in Fig. 2c. In addition, as discussed Fig. 4d, the energy shift $\Delta E_{\text{in-plane}}$ owing to the quantum resonance in the stacking direction is 22 and 20 meV in the QD multilayers prepared with low and high $\sigma_{\text{in-plane}}$, respectively. The difference between the absorption energies of isolated QD sample and 3D QDSL is 39 meV, which corresponds to the sum of the resonant coupling energies due to the quantum resonance in both the stacking and in-plane direction.

The shift of the absorption peak in the QDSL in which 1D quantum resonance occurs was more than that of the QDSL in which 2D quantum resonance occurs. The reason for this is that the in-plane QD distance of 3.9 nm is longer than the interlayer distance of 3.5 nm due to the electrostatic repulsion between the ligand of NAC. In addition, since there is no medium between the in-plane QDs and the medium between the QDs in the stacking direction is poly(diallyldimethylammonium chloride) (PDDA),

the presence of PDDA may make the quantum resonance between the QDs in the stacking direction stronger than in the in-plane direction.

**Dimensional effect of quantum resonance in QDSLs.** To evaluate the formation of the minibands in the QDSLs, we measured the excitation energy dependence of the PL spectra and detection energy dependence of the PLE spectra in the QDSL fabricated by the LBL method. For comparison, we performed the same measurements for the QD solution. The QDs are randomly dispersed in the solution, and sufficiently large distances between QDs cause no interactions between them. Figure 6a shows the excitation energy dependence of the PL spectra of the CdTe QD solution. Evidently, the PL peak shifts towards the lower energies as the excitation energy decreases, which corresponds to typical size-selective spectroscopy results[41–43]. The optical spectra of semiconductor QDs are usually inhomogeneously broadened because of the QD size distribution. When the excitation energy is low, relatively large QDs are selectively excited, and the PL peak energy decreases, and vice versa. The current experimental results observed in the QD solution originated from such selective excitation of a specific size of QDs that resonated with a specific excitation energy. In contrast, the experimental results for the QDSLs were completely different from those of the solution. Figure 6b–d show the excitation energy dependence of the PL spectra of the QDSLs in which the 1D, 2D, and 3D quantum resonance occurs, respectively. We found that the PL peak did not shift in any of the three QDSLs even when the excitation energy was changed; the size selectivity was not observed in the QDSLs.

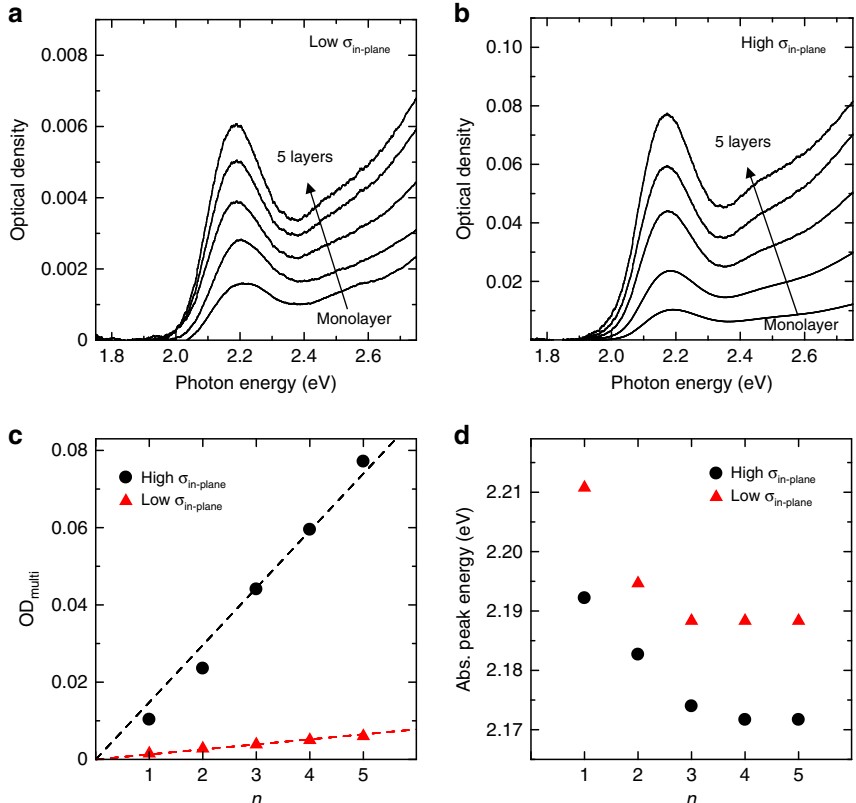

**Fig. 4 Absorption properties of quantum dot multilayers. a**, **b** Absorption spectra of the CdTe quantum dot multilayers with **a** low and **b** high in-plane quantum dot density ($\sigma_{\text{in-plane}}$). **c** Optical density of the absorption peak of the multilayers ($OD_{\text{multi}}$) versus the number of layers ($n$). The dashed lines are proportional to $n$, which show that the $OD_{\text{multi}}$ are proportional to $n$. **d** Absorption peak energy of the multilayers versus $n$.

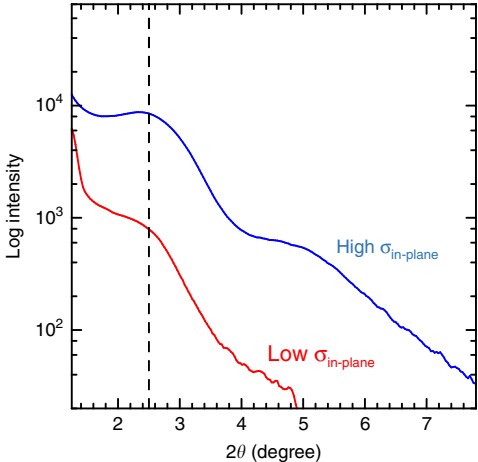

**Fig. 5 X-ray diffraction structural analysis of quantum dot multilayers.** Out-of-plane X-ray diffraction patterns of CdTe quantum dot 30 layers with low (red) and high (blue) in-plane quantum dot density ($\sigma_{\text{in-plane}}$). The dashed lines show the peak value of the X-ray diffraction.

We also measured the detection energy dependence of the PLE spectra in the QD solution and three QDSLs, as shown in Fig. 7. The inset represents the PL spectrum, and the down arrows indicate the detection energies. Similar to the PL spectra behaviour, the PLE peak energy increases as the detection energy increases in the case of the QD solution. This is again a typical result of the size-selective spectroscopy[41–43]. Figure 7b–d show the detection energy dependence of the PLE spectra in the three QDSLs in which the 1D, 2D, and 3D quantum resonance appears,

respectively. We found that the PLE peak energy did not shift even when the detection energy changed in all the QDSLs.

Thus far, Kagan and Crooker et al. have investigated the optical properties of CdSe and CdSe/ZnS QD solids and interpreted their results by considering the energy transfer (ET) mechanism[44–47]. In these QD solids, the shift of the absorption peak was not observed compared with solution samples, and thus it is considered that quantum resonance between QDs does not occur in these systems. Miyazaki et al. also discussed exciton hopping based on ET that occurs in QD arrays by comparing the optical properties of CdSe/ZnS QD arrays and solution samples[20,43]. No shift of absorption peak was observed in the CdSe/ZnS QD arrays as well[20]. That is, the quantum resonance between QDs did not occur, and only long-range ET between QDs occurred in the CdSe/ZnS QD arrays because the distance between the QDs is large in the CdSe/ZnS QDSLs owing to the presence of long ligands of octadecylamine. Furthermore, they reported experimental results on the excitation energy dependence of PL spectra in the QDSLs of octadecylamine-capped CdSe/ZnS QDs[43]. Although the shifts of the PL energy in the CdSe/ZnS QDSLs with respect to changes in the excitation energy were slightly smaller than that in solution samples, the size selectivity was still clearly observed even in the QDSLs.

To confirm this observation, we prepared OA-capped CdTe QDs and their QDSLs possessing a longer inter-QD distance (Supplementary Fig. 5a) and investigated the excitation energy dependence of the PL spectra. From Supplementary Fig. 5b, c, it can be seen that the PL peak shifts to the higher energy side as the excitation energy increases not only in the solution sample but also in QDSL, that is, the size selectivity is clearly observed. As shown in Supplementary Fig. 5d, in OA-capped CdTe QDSL, the shift of the PLE peak is also clearly observed when the detection

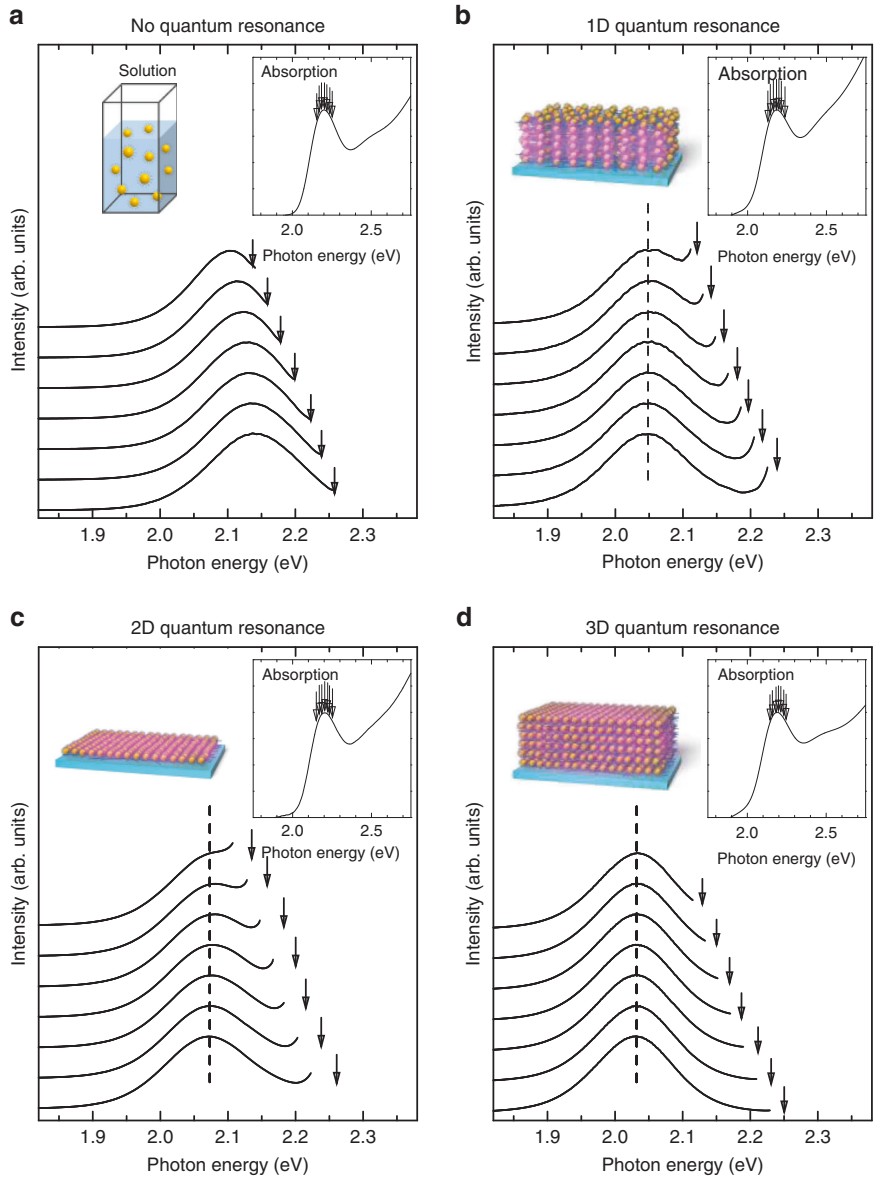

**Fig. 6 Excitation energy dependence of the photoluminescence spectra. a–d** Excitation energy dependence of the photoluminescence spectra of CdTe quantum dot solution in which quantum dots are randomly dispersed **a**, and CdTe quantum dot superlattices exhibiting 1D **b**, 2D **c**, and 3D **d** quantum resonances. The solid lines show photoluminescence spectra with different excitation energies. The excitation energy increases from top to bottom. The insets show the absorption spectra of each sample. The down arrows indicate the excitation energies in the photoluminescence measurements. The dashed lines show the peak value of the photoluminescence spectra.

energy is changed. In contrast, in the present NAC-capped CdTe QDSLs, the quantum resonance clearly occurs as discussed above, and the size selectivity is lost, reflecting the formation of the coupled states between QDs.

Thus far, in one-dimensional systems such as quantum wires and 1D QDSLs of InAs, it has been reported theoretically and experimentally that the radiative recombination lifetime depends on temperature and is proportional to $T^{0.5}$ [48–51]. We measured the temperature dependence of the PL-decay profiles to confirm that a one-dimensional miniband was formed in the QD multilayer prepared with the condition of low $\sigma_{\text{in-plane}}$. Supplementary Fig. 6a shows the PL-decay profiles measured at 80, 180, and 250 K. The solid curves are the results of fittings performed using three exponential functions. The temperature dependence of the average decay time is shown in Supplementary Fig. 6b. In the temperature range from 80 to 250 K, the PL intensity is almost constant, and that non-radiative recombination process

contributes minimally. Thus, the observed PL decay time reflects the radiative recombination lifetime. For the film samples in which CdTe QDs are randomly dispersed, the PL decay time is independent of temperature[52], while for the QD multilayer prepared with the condition of low $\sigma_{\text{in-plane}}$, the decay time depends on the temperature. Furthermore, in the temperature region above 80 K, the decay is proportional to $T^{0.5}$, which clearly demonstrates the formation of a one-dimensional miniband. In addition, we measured the temperature dependence of the PL decay times in the QDSLs in which 2D and 3D quantum resonances occur (Supplementary Fig. 6c, d) as well as 1D (Supplementary Fig. 6b). As shown in Supplementary Fig. 6c, d. PL decay times in CdTe QDSLs in which the 1D and 2D quantum resonances occur are proportional to $T^{0.5}$ and $T^{1.0}$, respectively. In ref. [49], Akiyama et al. theoretically and experimentally demonstrated that the PL decay times of quantum wires and quantum wells, in which 1D and 2D electronic states are formed,

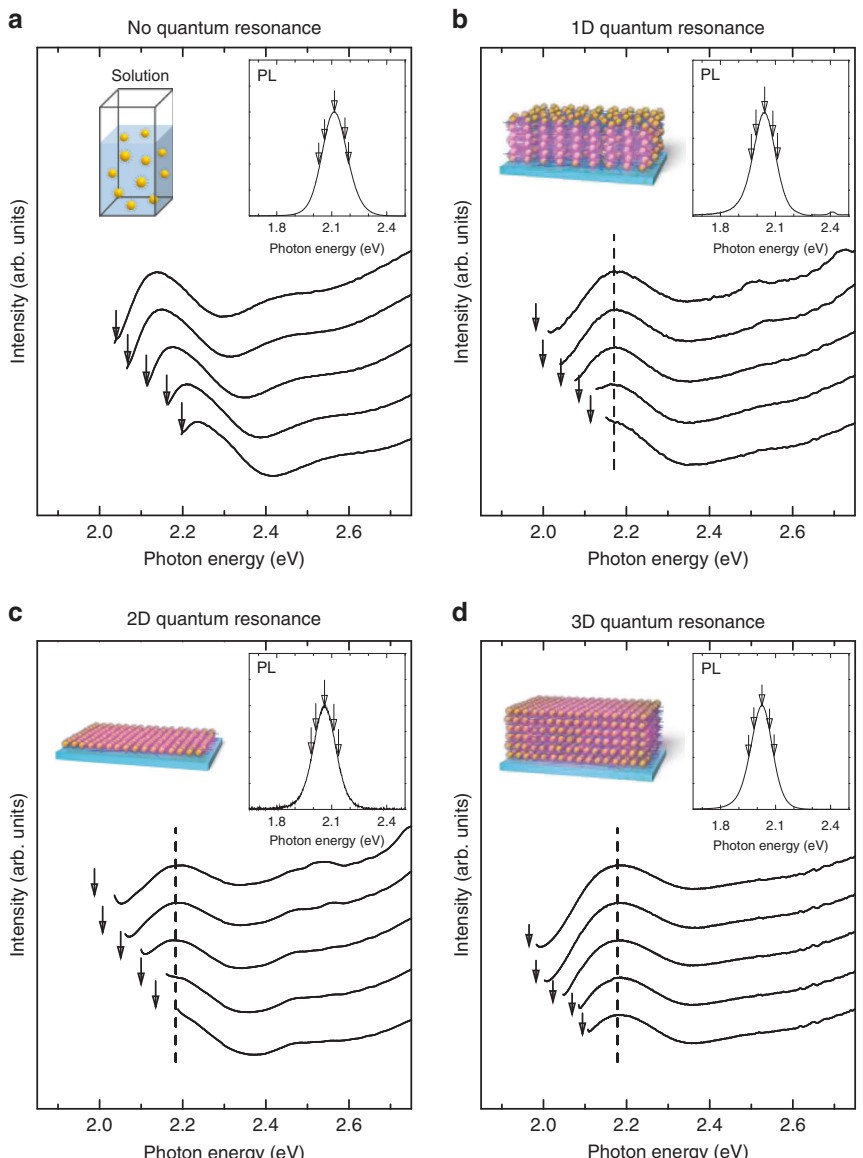

**Fig. 7 Detection energy dependence of the photoluminescence (PL) excitation spectra. a–d** Detection energy dependence of the PL excitation spectra of the CdTe quantum dot solution **a** and CdTe quantum dot superlattices in which the 1D **b**, 2D **c**, and 3D **d** quantum resonances occur. The solid lines show PL excitation spectra with different detection energies. The detection energy increases from top to bottom. The insets show the PL spectra of each sample. The down arrows indicate the detection energies in the PL excitation measurements. The dashed lines show the peak value of the PL excitation spectra.

are proportional to $T^{0.5}$ and $T^{1.0}$, respectively. Therefore, the present temperature dependence of PL decay times strongly demonstrate the formation of the 1D and 2D minibands in the CdTe QDSLs. Furthermore, the PL decay time in the 3D CdTe QDSLs is approximately proportional to $T^{1.5}$, demonstrating 3D miniband formation. We believe that these experimental results further demonstrate the quantum resonance-based miniband formation in the CdTe QDSLs.

The PL QYs for 1D, 2D, and 3D structures are 27%, 26%, and 22%, respectively, and the PL decay times are 2.7, 3.6, and 4.4 ns, respectively. The PL QY decreases slightly, and the PL decay time increases slightly as the dimensionality increases. These results suggest that excitons can be more widely delocalised owing to the stronger quantum resonance with the higher dimensionality. It is expected that the radiation mechanism in QDSLs will be further clarified by examining the transient absorption and its temperature dependence in detail.

## Discussion

We fabricated the CdTe QDSLs by LBL method and successfully controlled their dimensions of quantum resonance. The experimental results of the excitation energy dependence of the PL spectra and detection energy dependence of the PLE spectra clearly demonstrate the formation of coupled electronic states (minibands) in the CdTe QDSLs. This is the first report to show that the dimension of the quantum resonance can be controlled and that the resulting minibands are clearly formed. The fabrication of the CdTe QDSLs by the LBL method proposed in this study can be applied to other water-soluble semiconductor QDs and metal nanoparticles, as well as oxide, dielectric, and magnetic nanoparticles. Furthermore, the ability to combine different types of semiconductor QDs or combine semiconductor QDs with other nanoparticles will expand the possibilities of new material design including more complex heteronanostructures. QDSLs have been theoretically shown to have a higher multiple-exciton

generation efficiency than isolated QDs[53], and the efficiency changes depending on the dimension of QDSLs[54]. In this study, we were able to clarify the formation of the minibands based on the picture of the 1D, 2D, and 3D quantum resonance in the CdTe QDSLs. Further elucidation of novel optical properties such as the multiple-exciton generation and hot electron dynamics in QDSLs is desired.

## Methods

We prepared NAC-capped CdTe QDs using a previously reported procedure[34]. First, we added 250 mg of tellurium powder (purity 99.99%; purchased from Kojundo Chemical) and 312.5 mg of sodium borohydride (purity 98%; purchased from Kanto Chemical) to a vial and then added 6.0 mL of deionized (DI) water before sealing the vial. We then cooled the vial in an ice bath and allowed it to react for ~8 h. In the solution, the chemical reaction generates NaHTe as a tellurium ion source and $Na_2B_4O_7$ as a precipitate. Next, we dissolved NAC (purity 98%; purchased from Kishida Chemical) and $Cd(ClO_4)_2 \cdot 6H_2O$ (purity 99%; purchased from FUJIFILM Wako Pure Chemical) in 100 mL of DI water. We adjusted the solution to pH 7.0 via the stepwise addition of 0.2 mM NaOH (purchased from Kishida Chemical), and then, we added the prepared NaHTe solution. Finally, we added dilute HCl (purchased from Kishida Chemical) and adjusted the pH to 5.0 to complete the CdTe QD precursor. The molar ratio in the precursor was Cd:Te:NAC = 1.0:0.3:1.2, and the concentration was $[Cd^{2+}] = 20$ mM. We synthesised CdTe QDs by heating 10 mL of the precursor solution in an autoclave at 200 °C for 20 min and then cooling it in an ice bath.

We fabricated CdTe QDSLs via LBL assembly. First, we immersed a quartz substrate in a piranha solution (98% $H_2SO_4$:30% $H_2O_2$ = 7:3, v/v) heated at 150 °C to perform a hydrophilic treatment and to negatively charge the substrate surface. Next, we alternately immersed the substrate in a PDDA solution (20 wt % in $H_2O$; purchased from Sigma-Aldrich), which is a cationic polymer, and a polyacrylic acid (PAA) solution ((~25%); purchased from FUJIFILM Wako Pure Chemical), which is an anionic polymer. In this process, we washed the substrate with DI water before immersing the substrate in a different solution. After we fabricated the PDDA/(PAA/PDDA)$_2$ layers on the substrate, we immersed the substrate in a NAC-capped CdTe QD solution. Because NAC has negative chargeability in water, the QD monolayer can be fabricated on the substrate by utilising electrostatic interactions. In addition, we fabricated a QD multilayer by alternatively immersing the substrate in the PDDA solution and CdTe QD solution.

For the STEM and TEM measurements, we fabricated CdTe QD monolayers with low and high $\sigma_{in\text{-}plane}$ on a Cu grid supported by an elastic carbon film (ELS-C10 Cu100P grid; purchased from OkenShoji). First, we hydrophilized the surface of the grid using a plasma treatment. Next, we prepared QD monolayers by drop-casting the PDDA, PAA, and QD solutions onto the grid in the same order as described previously. During this process, we washed the grid with DI water before the different solutions were drop-cast.

OA-capped CdTe was synthesised by hot-injection methods[55]. In a glovebox, a Te solution was prepared by dissolving 0.0512 g of Te powder (0.40 mmol) in 0.414 g of tri-$n$-butylphosphine (TBP) and subsequent dilution with 19.94 mL of 1-octadecene (ODE). A Cd source mixture was prepared by dissolving 0.0128 g of CdO (0.10 mmol), 0.1138 mL of OA (0.36 mmol), and 4.91 mL of ODE at 295 °C in a three-neck flask to obtain a clear solution. At this temperature, 2.535 mL of Te (0.05 mmol) solution was quickly injected into the Cd source solution. After 1 min from the injection, reaction vessel was transferred into ice bath to stop the reaction. For the purification, 4 mL of as-prepared CdTe QD solution and a mixture of hexane and methanol (1/2, v/v) were vigorously shook in a 15 mL centrifuge tube and then centrifuged at 10,000 rpm for 5 min. After the ODE phase was extracted, same procedure was repeated once and then the CdTe QD in the ODE phase was precipitated with 40 mL of acetone. The OA-capped CdTe QD solution was obtained by centrifugation (10,000 rpm for 10 min), decantation and re-dispersion into 4 mL of toluene. All synthesis and purification processes were carried out under nitrogen flow condition.

We recorded the extinction spectra using a JASCO V-650 spectrometer with a spectral resolution of 0.2 nm. The absorption spectra specific to CdTe QD layers were obtained by correcting the baseline in the measured extinction spectra. The details of the analysis of the absorption spectra is described in Supplementary Information. We measured the PL and PLE spectra using a JASCO FP-8300 spectrofluorometer with a spectral resolution of 0.5 nm. The PL QYs were measured using a JASCO FP-8300 spectrofluorometer equipped with an integrating sphere. For the measurements of PL-decay profiles, a laser-diode (405 nm, PicoQuant LDH-P-C-405) with a pulse duration of 50 ps and a repetition of 125 kHz was used as the excitation light. The PL-decay profiles were recorded using a Hamamatsu C5094 imaging spectrograph and C4334 streak scope. We recorded XRD patterns with a Rigaku SmartLab using Cu Kα radiation ($\lambda = 0.154$ nm). We performed STEM measurements by using a Thermo Fisher Scientific Talos at an accelerating voltage of 200 kV, and we obtained TEM images using a JEOL JEM-2100F/SP operated at 200 kV.

## Data availability

The data that support the findings of this study are available from the corresponding author upon reasonable request.

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

## Acknowledgements

The authors acknowledge Thermo Fisher Co. for the STEM measurements. K.H.-D. acknowledges financial support from JST (PRESTO), Toyota Mobility Foundation, and Grant-in-Aid for Scientific Research on Innovative Areas, Grant No. 18H05407. D.K. acknowledges financial support from JSPS KAKENHI, Grant Nos. 24560015 and 20H02549.

## Author contributions

T.L. and K.O. designed the experiment, prepared CdTe QDs and QDSLs, acquired and analysed absorption, PL, and PLE spectra. K.E. acquired TEM images, prepared OA-capped CdTe QDSLs, and analysed PL and PLE spectra. D.I. and T.K. supported the STEM measurements. K.H.-D., Y.-J.P. and D.K. provided conceptual advice. All authors contributed to the interpretation of the results and preparation of the manuscript.

## Competing interests

The authors declare no competing interests.
