## [Peer Review File · Nature Communications]

Reviewers' comments:

Reviewer #1 (Remarks to the Author):

The authors aim to control the degree of quantum resonance by engineering in-plane and out-of-plane QD assembly. The authors seemed to intentionally ignore the mechanism of energy transfer, such as FRET, between QDs, which is well established. The formation of miniband itself is of importance in transport-based devices, but the authors did not discuss any transport properties of the resulting films. In fact, the formation of miniband should be rigorously verified by measuring the transport characteristics at various low temperatures. The degree of absorption redshift upon QD assembly cannot unambiguously corroborate quantum resonance, in particular other complications, such as FRET, also come into play.

With the above reasons in mind, overall I do not find the current form of this manuscript is sufficient to be published in Nature Comm. I have the following minor questions that the authors should consider.

1. The authors stated in the intro that when the spacer is greater than 2nm, there is no out of plane quantum resonance. Why? The FRET distance can be as long as 10 nm.
2. The authors should also include the XRD spectrum for the low -coverage monolayer sample (Fig. 2d). If the required angle is too small, the authors should do small-angle XRD/GIXD.
3. Fig. 5 did not see the spacer distance. Why?
4. Fig. 6 can be easily explained with the FRET mechanism. Regardless of excitation wavelength, the emission all comes from the larger QDs due to the FRET mechanism. It is not about quantum resonance.
5. More spectroscopic results, such as excitation-dependent QY and TRPL should be used to exclude the mechanism of FRET.

Reviewer #2 (Remarks to the Author):

The authors construct assemblies of CdTe that they claim have tunable mini-bands, where they can realize band-like properties in either the xy or z directions, or both. Although the assemblies are very nice, I would recommend rejecting this paper, both because it lacks novelty and because there are several serious scientific issues:

- 1) Introduction is not written very carefully: (i) QDs are not "new" fluorescent materials for any application; (ii) it is not true, in general, that MEG effectively occurs in QDs (plus it's not relevant to this study); (iii) going all the way back to a description of the original hot injection synthesis is totally superfluous for this paper; (iv) the claim that mini-band formation has not been discussed in detail is certainly not true. Even a simple Google search reveals at least tens of papers on the photophysical aspects of this topic.
- 2) all of the ligands that the authors mention as water-solubilizing ligands in the intro bind through thiolates, which is a PL-quencher for almost all types of metal-chalcogenide QDs, so mentioning these ligands as a strategy to avoid PL-quenching exciton-delocalizing ligands (in the previous paragraph) makes no sense. Also, there are thiolate exciton-delocalizing ligands (described in papers that the authors did not cite).
- 3) I'm not convinced at all by the most important datasets of this paper, the absorption spectra as a function of OD. I think the linewidth of the absorption peak is a better indicator of conjugation among QDs than the peak position. First of all, the peak appears to be broadening with increased monolayer OD, which is not reported here. Second, as the peak broadens it becomes more difficult to pick the peak without a fit (which I also don't see here).
- 4) The authors claim that 0.5 nm is short enough to realize mini-band formation. Where does this number

come from? Where is the evidence for it? It's completely arbitrary.

5) Similarly, there is not nearly enough evidence or precision in the absorbance spectra to make the quantitative claim that 19 meV is the coupling energy among QDs.

6) the STEM images shown in Figure 3 don't show anywhere near sub-nanometer control of interparticle distance that the authors claim is necessary to tune mini-band formation. The images either show a sub-monolayer or a monolayer. There isn't even enough resolution in the images to determine sub-nanometer distances.

7) page 5: i don't see any difference in the amount of shift in abs peak between Figures 4a and 4b. The authors need to be clear about how they got their plots in 4c,d. This is subject to a lot of error, without a fit. Also, i would expect a significant baseline for sub-monolayer coverage. Were these spectra collected with an integration sphere? Where is the baseline? Is it subtracted? The subtraction procedure can shift the apparent position of the excitonic peak.

8) I thought the PLE experiments were supposed to differentiate between the different quantum resonances (1D, 2D, 3D), but they all look the same, so why plot them all?

Reviewer #3 (Remarks to the Author):

In their manuscript, the authors show the fabrication and optical experiments with superlattices of colloidal CdTe quantum dots. In addition to in-plane 2D assembly, layer-by-layer deposition allows very controlled fabrication of 1D and 3D structures. The results provide evidence that inter-dot electronic coupling establishes a "quantum resonance" shifted by 19 meV to 39 meV from the isolated QD emission, dependent on the dimensionality. With this degree of control it goes significantly beyond what is there in the literature and is likely to excite a broader audience, especially of course in the colloidal quantum dot community where tailored superlattices have raised increasing attention in recent years.

Overall, the paper is a pleasure to read, and the fabrication and experiments appear very carefully and diligently crafted to yield solid results that are corroborated also by contrasting them to control experiments with isolated QDs. Furthermore, the used methods are well described with sufficient detail, and the samples are thoroughly characterized. Nevertheless, I believe that a small number of points should be addressed before the manuscript is published in Nature Communications.

1) Can the authors explain why the energy shift from the quantum resonance going from isolated QDs to 1D is even slightly larger than going from isolated QDs to 2D, i.e. 22 meV vs. 19 meV respectively (Fig. S2). Is the small difference in periodicity (3.9 nm for in-plane and 3.5 nm for out-of-plane structures) sufficient, as the scattering peaks (Fig. 2d and Fig. 5) and also the distributions obtained by STEM (Fig. 3d) are rather broad? Because in principle, I would expect otherwise, i.e. that for the 2D structures the shift should be much larger than the 1D shift.

2) While the lifetime vs. temperature dependence of 1D quantum resonance structures show the expected $T^{0.5}$ behavior (Fig. S4) data for the 2D and 3D cases are not shown or mentioned. Ref. 53 suggests e.g. that the 2D case should show T^1 . Such data would substantiate further the claim on the achieved dimensionality.

3) Could a recent model based on altering the dielectric confinement instead of quantum tunneling (arXiv 1912.04182) provide an alternative explanation for the findings?

4) For the low-density in-plane structures that are used for the individual layers in the 1D case, it should be discussed what material actually fills the voids between the QDs. Furthermore, the sketch for the isolated and 1D case in Fig. 1 suggests that there could also be some in-plane periodicity established, while the data

in Fig. 3a/c clearly show that this is not the case. It should be possible to improve Fig. 1 to reflect better the lateral randomness.

5) In many of the graphs there are dashed lines to highlight certain features or represent a fit to the data (Fig. 2b, 2d, 4c, 5, 6b-d, 7b-d). It should be briefly described in the respective figure captions.

Dear Dr. Selina La Barbera
Associate Editor, Nature Communications

The manuscript was rechecked and the necessary changes have been made in accordance with the reviewers' suggestions. The responses to all the comments are given in this letter.

We would like to emphasize the novelty of this study. As Reviewer #2 pointed out, the formation of the miniband in the quantum dot (QD) solids has been discussed mainly from the viewpoint of transport properties. For the photophysical properties of QD solids, the absorption properties have been mainly investigated, and the quantum resonance have been discussed only from the viewpoint of the low-energy shift or the broadening of absorption peaks.

In this study, the formation of minibands derived from the quantum resonance between QDs is investigated by systematically measuring the excitation energy dependence of the photoluminescence (PL) spectra, the detection energy dependence of PL excitation spectra, and the temperature dependence of the PL decay time, as well as the low energy shift of the absorption peak. In this revision, to further prove the formation of miniband by the quantum resonance, we additionally measured the temperature dependence of the dimension-dependent PL decay time in the QD superlattices in which 1D, 2D, and 3D quantum resonances occur. We also responded to all concerns of the reviewers in detail and revised the manuscript according to the reviewer's suggestions.

We note that these results are significant in suggesting that the formation of minibands can be directly observed by experiments other than the transport properties. We believe that this is the clear novelty of this study.

We hope that the revised manuscript is suitable for publication.

Sincerely yours,

D. Kim
Department of Applied Physics, Osaka City University
Osaka 558-8585, Japan
TEL&FAX: +81-6-6605-3087
E-mail: tegi@a-phys.eng.osaka-cu.ac.jp

Reply to Reviewer 1

The authors aim to control the degree of quantum resonance by engineering in-plane and out-of-plane QD assembly. The authors seemed to intentionally ignore the mechanism of energy transfer, such as FRET, between QDs, which is well established. The formation of miniband itself is of importance in transport-based devices, but the authors did not discuss any transport properties of the resulting films. In fact, the formation of miniband should be rigorously verified by measuring the transport characteristics at various low temperatures. The degree of absorption redshift upon QD assembly cannot unambiguously corroborate quantum resonance, in particular other complications, such as FRET, also come into play. With the above reasons in mind, overall I do not find the current form of this manuscript is sufficient to be published in Nature Comm. I have the following minor questions that the authors should consider.

Our reply: Thank you very much for your detailed review of our paper. It is very helpful to improve the paper. We carefully addressed all of your comments and questions. The one-to-one replies are written below. References listed in the revised manuscript are represented by numbers. The revised parts of the manuscript are highlighted as red characters in the revised manuscript.

Thus far, Kagan and Crooker et al. have investigated the optical properties of CdSe and CdSe/ZnS QD solids and interpreted their results by considering the energy transfer (ET) mechanism [44-47]. In these QD solids, the shift of the absorption peak was not observed compared with solution samples, and thus, it is considered that the quantum resonance between QDs does not occur in these systems. Miyazaki et al. also discussed exciton hopping that occurs in QD arrays by comparing the optical properties of CdSe/ZnS QD arrays and solution samples [20,43]. No shift of absorption peak was observed in the CdSe/ZnS QD arrays as well (See Fig. 2 of ref. 20). That is, the quantum resonance between QDs did not occur, and only long-range ET occurred between QDs in the CdSe/ZnS QD arrays. In the samples where the short-range quantum resonance does not occur (i.e., coupled states are not formed), a clear shift is observed in the excitation energy dependence of the photoluminescence (PL) spectra (See Fig. 1 of ref. 43). In this study, we also prepared QD superlattices (SLs) of oleic acid (OA)-capped CdTe possessing a longer inter-QD distance and measured the excitation energy dependence of the PL spectra. The results are shown in Fig. S5. In QDSLs of OA-capped CdTe QDs, a clear shift of PL peak is observed.

On the other hand, in the *N*-acetyl-*L*-cysteine (NAC)-capped CdTe QDSLs proposed in this study, a low energy shift of the absorption peak is clearly observed, and there is no shift in the excitation energy dependence of the PL spectra and the detection energy dependence of the PL excitation (PLE) spectra (See Figs. 6 and 7). These results are considered to be clear experimental evidences that the coupled state is formed by the short-range quantum resonance. Therefore, we do not insist

on the formation of the coupled state solely from the low energy shift of the absorption peak any more.

As the reviewer pointed out, the formation of minibands in CdSe [13-15], InAs [16], and PbSe QD solids [17,18] has been discussed mainly from the viewpoint of transport properties. In contrast, in this study, the formation of minibands derived from the short-range quantum resonance between QDs can be discussed from the viewpoint of optical properties such as the excitation energy dependence of the PL spectra (Fig. 6), the detection energy dependence of PLE spectra (Fig. 7), and the temperature dependence of the PL decay time (Fig. S6), as well as the low energy shift of the absorption peak. These results are significant in suggesting that the formation of minibands can be directly observed by experiments other than the transport properties. We believe that this is the clear novelty of this study.

We added the above discussion on page 2, line 32–page 3, line 2 and page 7, lines 19–31 in the revised manuscript, and the following references are also newly added.

- [13] Kovalenko, M. V., Scheele, M. & Talapin, D. V. Colloidal nanocrystals with molecular metal chalcogenide surface ligands. *Science* **324**, 1417 (2009).
- [16] Jang, J., Liu, W., Son, J. S. and Talapin, D. V. Temperature-dependent hall and field-effect mobility in strongly coupled all-inorganic nanocrystal arrays. *Nano Lett.* **14**, 653-662 (2014).
- [44] Kagan, C. R., Murray, C. B. & Bawendi, M. G. Long-range resonance transfer of electronic excitations in close-packed CdSe quantum-dot solids. *Phys. Rev. B* **54**, 8633 (1996).
- [45] Kagan, C. R., Murray, C. B., Nirmal, M. & Bawendi, M. G. Electronic energy transfer in CdSe quantum dot solids. *Phys. Rev. B* **76**, 1517 (1996).
- [46] Crooker, S. A., Hollingsworth, J. A., Tretiak, S. & V. I. Klimov Spectrally resolved dynamics of energy transfer in quantum-dot assemblies: Towards engineered energy flows in artificial materials. *Phys. Rev. Lett.* **89**, 186802 (2002).
- [47] Achermann, M., Petruska, M. A., Crooker, S. A. & Klimov, V. I. Picosecond energy transfer in quantum dot Langmuir-Blodgett nanoassemblies. *J. Phys. Chem. B* **107**, 13782-13787 (2003).

1. The authors stated in the intro that when the spacer is greater than 2nm, there is no out of plane quantum resonance. Why? The FRET distance can be as long as 10 nm.

Our reply: As the reviewer pointed out, the FRET occurs even if the centre-to-centre distance between QDs is more than 10 nm, since the ET is a long-range interaction based on the dipole-dipole interaction. In the previous work [40], we also experimentally verified the ET mechanism from the distance dependence of the ET rate.

In contrast, the quantum resonance arises from the coupling of the wave functions between the adjacent QDs, which is a short-range interaction, and its strength weakens exponentially with the surface-to-surface distance between QDs. In the previous work [11], we observed the experimental results that the shift amount of the absorption peak energy decreases exponentially with QD

interlayer distance (See Fig. 3a in ref. 11). This clearly demonstrates that the observed absorption peak shift is due to the short-range coupling of wave functions, that is, the quantum resonance. When the QD interlayer distance is larger than 2 nm, the absorption peak hardly shifts (See Fig. 3a in ref. 11). Therefore, it can be concluded that the quantum resonance hardly occurs when the QD distance is approximately 2 nm or more. It should be also mentioned that the threshold distance changes more or less depending on the QD composition, diameter, and type of ligand.

We added the above discussion on page 4, lines 15–20 in the revised manuscript.

2. The authors should also include the XRD spectrum for the low-coverage monolayer sample (Fig. 2d). If the required angle is too small, the authors should do small-angle XRD/GIXD.

Our reply: The average centre-to-centre distance estimated from STEM image of QD monolayer with the low in-plane QD density ($\sigma_{\text{in-plane}}$) is 5.6 nm. Thus, it is expected that the XRD peak will be observed at 2θ angle of 1.6° , which is within the measurable range under the experimental conditions in this study. However, no peak was observed in the in-plane XRD measurement of the QD monolayer with the low $\sigma_{\text{in-plane}}$. There are two possible reasons: (1) XRD peak intensity decreases as QD density decreases, and (2) XRD signals broadens due to the random arrangement in the QD monolayer with the low $\sigma_{\text{in-plane}}$. From the STEM images in Figs. 3a and 3b, it can be concluded that the average distance between QDs is clearly different. The coincidence of the inter-QD distance estimated from the in-plane XRD and STEM in the QD monolayer with high $\sigma_{\text{in-plane}}$ also ensures the credibility of the inter-QD distance.

3. Fig. 5 did not see the spacer distance. Why?

Our reply: The average QD diameter is estimated to be 3.4 nm from the TEM image shown in Fig. S1a, and the average distance in the stacking direction in the QD multilayers is estimated to be approximately 3.5 nm from the XRD results shown in Fig. 5. Therefore, the thickness of the PDDA layer is estimated to be approximately 0.1 nm. In contrast, as discussed in ref. 40, the average thickness of 1 bilayer of polycation of PDDA and polyanion of poly(acrylic acid) (PAA) was estimated to be 0.9 nm from the measurements of spectroscopic ellipsometry of multilayers of PDDA/PAA. Thus, the thickness of the PDDA monolayer is expected to be 0.4–0.5 nm, which is half of the thickness of the PDDA/PAA bilayer. It was difficult to observe the clear difference in the thickness of the PDDA layer because the XRD peak shown in Fig. 5 is broad due to the disorder of the regularity in the stacking direction. However, since diffraction peaks are observed at almost the same angle, we note that they have almost the same periodicity in the stacking direction.

4. Fig. 6 can be easily explained with the FRET mechanism. Regardless of excitation wavelength, the emission all comes from the larger QDs due to the FRET mechanism. It is not about quantum resonance.

Our reply: As mentioned at the beginning of the reply letter, in the QDSL where the quantum resonance between QDs does not occur and only ET between QDs is observed, clear shift in the excitation energy dependence of the PL spectra is experimentally observed [43]. To reconfirm this observation, we prepared the QDSLs of the OA-capped CdTe QDs possessing long inter-QD distance and measured the excitation energy dependence of the PL spectra. As shown in Fig. S5c, PL peak was shifted depending on the excitation energy. Thus, we conclude that experimental results in Fig. 6 cannot be explained with the FRET mechanism. In contrast, the peaks do not shift regardless of the excitation energy of the PL spectra and the detection energy of the PLE spectra in the present QDSLs of the NAC-capped CdTe QDs as shown in Figs. 6 and 7.

As will be stated in the reply to the reviewer's comment 5, to prove that the experimental results obtained in this study are not due to ET but due to the formation of miniband by the quantum resonance, we measured the temperature dependence of the PL decay time in the QDSLs in which 1D, 2D, and 3D quantum resonances occur (Figs. S6b, S6c, and S6d, respectively). As shown in Figs. S6b and S6c, PL decay times in CdTe QDSLs in which the 1D and 2D quantum resonances occur are proportional to $T^{0.5}$ and $T^{1.0}$, respectively. In ref. 49, Akiyama et al. theoretically and experimentally demonstrated that the PL decay times of quantum wires and quantum wells, in which 1D and 2D electronic states are formed, are proportional to $T^{0.5}$ and $T^{1.0}$, respectively. Therefore, the present experimental results of temperature dependence of PL decay time strongly demonstrate the formation of the 1D and 2D minibands in the CdTe QDSLs. Furthermore, the PL decay time in the 3D CdTe QDSLs is approximately proportional to $T^{1.5}$, also demonstrating 3D miniband formation.

In this study, by systematically investigating the excitation energy dependence of the PL spectra (Fig. 6), the detection energy dependence of PLE spectra (Fig. 7), and the temperature dependence of the PL decay time (Fig. S6), as well as the low energy shift of the absorption peak, we concluded the formation of miniband derived from the quantum resonance.

We added the above discussion on page 8, lines 18–29 in the revised manuscript and page 2, line 32–page 3, line 3 in the revised supplementary information.

Revised Figure S6

Supplementary Figure S6 | Temperature dependence of radiative lifetime in the QDSL in which 1D, 2D, and 3D quantum resonance occurs. **a**, The dotted lines are PL-decay profiles measured at 80 K (red), 180 K (blue), 250 K (black), respectively. The solid curves are the results of fitting performed using three exponential functions. **b**, The open circles represent the average decay times at each of the temperatures of the QD layer in which 1D quantum resonance occurs; the solid line represents a $T^{0.5}$ dependence. **c**, The open rectangles represent the average decay times at each of the temperatures of the QD layer in which 2D quantum resonance occurs; the solid line represents a $T^{1.0}$ dependence, and dashed line represents a $T^{0.5}$ dependence. **d**, The triangles represent the average decay times at each of the temperatures of the QD layer in which 3D quantum resonance occurs; the solid line represents a $T^{1.5}$ dependence, and dashed line represents a $T^{0.5}$ dependence.

5. More spectroscopic results, such as excitation dependent QY and TRPL should be used to exclude the mechanism of FRET.

Our reply: Even if the FRET or the quantum resonance occurs in the QDSLs, their QY is almost independent of the excitation wavelength. Therefore, it is difficult to distinguish the quantum

resonance from FRET by the results of the excitation dependent QY. The TRPL measurement, which is another measurement proposed by the reviewer, has been used to investigate the FRET-based exciton hopping process in the QDSLs [43]. Since the carrier relaxation takes place due to the formation of minibands in the present NAC-capped CdTe QDSLs, it is difficult to distinguish the two mechanisms by the TRPL measurements.

As mentioned in the above reply to the reviewer's comment 4, we additionally measured the temperature dependence of the PL decay time in the QDSLs in which 1D, 2D, and 3D quantum resonances occur (Figs. S6). As shown in Figs. S6b and S6c, PL decay times in CdTe QDSLs in which the 1D and 2D quantum resonances occur are proportional to $T^{0.5}$ and $T^{1.0}$, respectively. In ref. 49, Akiyama et al. theoretically and experimentally demonstrated that the PL decay times of quantum wires and quantum wells, in which 1D and 2D electronic states are formed, are proportional to $T^{0.5}$ and $T^{1.0}$, respectively. Therefore, the present temperature dependence of PL decay time strongly demonstrates the formation of the 1D and 2D minibands in the CdTe QDSLs. Furthermore, the PL decay time in the 3D CdTe QDSLs is approximately proportional to $T^{1.5}$, also demonstrating 3D miniband formation.

We believe that the formation of miniband derived from the quantum resonance can be clearly demonstrated by systematically investigating the excitation energy dependence of the PL spectra (Fig. 6), the detection energy dependence of PLE spectra (Fig. 7), and the temperature dependence of the PL decay time (Fig. S6), as well as the low energy shift of the absorption peak.

As mentioned at the reply to the reviewer's comment 4, we added the above discussion in the revised manuscript.

Reply to Reviewer 2

The authors construct assemblies of CdTe that they claim have tunable mini-bands, where they can realize band-like properties in either the xy or z directions, or both. Although the assemblies are very nice, I would recommend rejecting this paper, both because it lacks novelty and because there are several serious scientific issues.

Our reply: Thank you very much for your detailed review of our paper. It is very helpful to improve the paper. We carefully addressed all of your comments and questions. The one-to-one replies are written below. The references listed only in this reply letter are represented by alphabets. In addition, the revised parts of the manuscript are highlighted as red characters in the revised manuscript.

The formation of the miniband in the CdSe [13-15], InAs [16], and PbSe [17,18] QD solids has been discussed mainly from the viewpoint of transport properties. In contrast, the formation of minibands derived from the quantum resonance between QDs by our approach can be discussed from the viewpoint of optical properties such as the excitation energy dependence of the PL spectra (Figs. 6), the detection energy dependence of the PL excitation (PLE) spectra (Figs. 7).

Thus far, Kagan and Crooker et al. have investigated the optical properties of CdSe and CdSe/ZnS QD solids and interpreted their results by considering the ET mechanism [44-47]. In these QD solids, the shift of the absorption peak was not observed compared with solution samples, and thus, it is considered that the quantum resonance between QDs does not occur in these systems. Miyazaki et al. also discussed exciton hopping that occurs in QD arrays by comparing the optical properties of CdSe/ZnS QD arrays and solution samples [20,43]. No shift of absorption peak was observed in the CdSe/ZnS QD arrays as well (See Fig. 2 of ref. 20). That is, the quantum resonance between QDs did not occur, and only long-range ET occurred between QDs in the CdSe/ZnS QD arrays. In the samples where the short-range quantum resonance does not occur (i.e., coupled states are not formed), a clear shift is observed in the excitation energy dependence of the PL spectra (See Fig. 1 of ref. 43). In this study, we also prepared QDSLs of OA-capped CdTe possessing a longer inter-QD distance and measured the excitation energy dependence of the PL spectra. As shown in Fig. S5, a clear shift of PL peak is observed in the QDSLs of OA-capped CdTe QDs.

On the other hand, in the NAC-capped CdTe QDSLs proposed in this study, a low energy shift of the absorption peak is clearly observed, and there is no shift in the excitation energy dependence of the PL spectra and the detection energy dependence of the PLE spectra (See Figs. 6 and 7). These results are considered to be clear experimental evidences that the coupled state is formed by the short-range quantum resonance. Therefore, we do not insist on the formation of the coupled state solely from the low energy shift of the absorption peak any more.

Furthermore, to prove that the experimental results obtained in this study are due to the formation of miniband derived from the quantum resonance, we additionally measured the

temperature dependence of the PL decay time in the QDSLs in which 1D, 2D, and 3D quantum resonances occur (Figs. S6b, S6c, and S6d). As shown in Figs. S6b and S6c, PL decay times in CdTe QDSLs in which the 1D and 2D quantum resonances occur are proportional to $T^{0.5}$ and $T^{1.0}$, respectively. In ref. 49, Akiyama et al. theoretically and experimentally demonstrated that the PL decay times of quantum wires and quantum wells, in which 1D and 2D electronic states are formed, are proportional to $T^{0.5}$ and $T^{1.0}$, respectively. Therefore, the present temperature dependence of PL decay time strongly demonstrates the formation of the 1D and 2D minibands in the CdTe QDSLs. Furthermore, the PL decay time in the 3D CdTe QDSLs is approximately proportional to $T^{1.5}$, also demonstrating 3D miniband formation.

Therefore, in this study, by systematically investigating the excitation energy dependence of the PL spectra (Fig. 6), the detection energy dependence of the PLE spectra (Fig. 7), and the temperature dependence of the PL decay time (Fig. S6), as well as the low energy shift of the absorption peak, we concluded the formation of miniband derived from the quantum resonance. We believe that this is the clear novelty of this study.

We added the above discussion on page 2, line 32–page 3, line 2 and page 7, lines 19–31 in the revised manuscript. In addition, we added the discussion of the temperature dependence of the PL decay time on page 8, lines 18–29 in the revised manuscript and page 2, line 32–page 3, line 3 in the revised supplementary information. The following references are also newly added.

- [13] Kovalenko, M. V., Scheele, M. & Talapin, D. V. Colloidal nanocrystals with molecular metal chalcogenide surface ligands. *Science* **324**, 1417 (2009).
- [16] Jang, J., Liu, W., Son, J. S. and & Talapin, D. V. Temperature-dependent hall and field-effect mobility in strongly coupled all-inorganic nanocrystal arrays. *Nano Lett.* **14**, 653-662 (2014).
- [44] Kagan, C. R., Murray, C. B. & Bawendi, M. G. Long-range resonance transfer of electronic excitations in close-packed CdSe quantum-dot solids. *Phys. Rev. B* **54**, 8633 (1996).
- [45] Kagan, C. R., Murray, C. B., Nirmal, M. & Bawendi, M. G. Electronic energy transfer in CdSe quantum dot solids. *Phys. Rev. B* **76**, 1517 (1996).
- [46] Crooker, S. A., Hollingsworth, J. A., Tretiak, S. & V. I. Klimov Spectrally resolved dynamics of energy transfer in quantum-dot assemblies: Towards engineered energy flows in artificial materials. *Phys. Rev. Lett.* **89**, 186802 (2002).
- [47] Achermann, M., Petruska, M. A., Crooker, S. A. & Klimov, V. I. Picosecond energy transfer in quantum dot Langmuir-Blodgett nanoassemblies. *J. Phys. Chem. B* **107**, 13782-13787 (2003).

Revised Figure S6

Supplementary Figure S6 | Temperature dependence of radiative lifetime in the QDSL in which 1D, 2D, and 3D quantum resonance occurs. **a**, The dotted lines are PL-decay profiles measured at 80 K (red), 180 K (blue), 250 K (black), respectively. The solid curves are the results of fitting performed using three exponential functions. **b**, The open circles represent the average decay times at each of the temperatures of the QD layer in which 1D quantum resonance occurs; the solid line represents a $T^{0.5}$ dependence. **c**, The open rectangles represent the average decay times at each of the temperatures of the QD layer in which 2D quantum resonance occurs; the solid line represents a $T^{1.0}$ dependence, and dashed line represents a $T^{0.5}$ dependence. **d**, The triangles represent the average decay times at each of the temperatures of the QD layer in which 3D quantum resonance occurs; the solid line represents a $T^{1.5}$ dependence, and dashed line represents a $T^{0.5}$ dependence.

1) Introduction is not written very carefully: (i) QDs are not "new" fluorescent materials for any application; (ii) it is not true, in general, that MEG effectively occurs in QDs (plus it's not relevant to this study); (iii) going all the way back to a description of the original hot injection synthesis is totally superfluous for this paper; (iv) the claim that mini-band formation has not

been discussed in detail is certainly not true. Even a simple Google search reveals at least tens of papers on the photophysical aspects of this topic.

Our reply: (i) According to the reviewer's comment, we revised the sentence on page 1, lines 28–29 in the revised manuscript.

(ii) According to the reviewer's comment, we deleted the description about multiple-exciton generation (MEG) on page 1, lines 30–32 in the original manuscript.

(iii) According to the reviewer's comment, we deleted the description about hot injection synthesis on page 1, line 33–page 2, line 4 in the original manuscript.

(iv) As mentioned above, the quantum resonance between QDs and the formation of miniband in CdSe [13-15], InAs [16], and PbSe [17,18] QD solids have been discussed mainly from the results of the transport properties. For the photophysical properties of QD solids, the absorption properties have been mainly investigated, and the quantum resonance have been discussed from the observation of the low-energy shift or the broadening of absorption peaks [13-18],[a-c]. In contrast, in this study, we investigated the quantum resonance by systematically measuring the excitation energy dependence of the PL spectra, the detection energy dependence of PLE spectra, and the temperature dependence of the PL decay time as well as the low-energy shift of the absorption peak. These results are significant in suggesting that the formation of minibands can be directly observed by experiments other than transport. We believe that this is the clear novelty of this study.

We added the above discussion on page 2, line 32–page 3, line 2 in the revised manuscript.

References

- [a] Williams, K. J., Tisdale, W. A., Leschkies, K. S., Haugstad, G., Norris, D. J., Aydil, E. S. & Zhu, X.-Y. Strong electronic coupling in two-dimensional assemblies of colloidal PbSe Quantum Dots. *ACSNano* **3**, 1532-1538 (2009).
- [b] Gao, Y., Talgorn, E., Aerts, M., Trinh, M. T., Schins, J. M., Houtepen, A. J. & Siebbeles L. D. A. Enhanced hot-carrier cooling and ultrafast spectral diffusion in strongly coupled PbSe quantum-dot solids. *Nano Lett.* **11**, 5471-5476 (2011).
- [c] Jazi, M. A., Janssen, V. A. E. C., Evers, W. H., Tadjine, A., Delerue, C., Siebbeles, L. D. A., H., van der Zant, H. S. J., Houtepen, A. J. & Vanmaekelbergh, D. Transport properties of a two-dimensional PbSe square superstructure in an electrolyte-gated transistor. *Nano Lett.* **17**, 5238-5243 (2017).

2) all of the ligands that the authors mention as water-solubilizing ligands in the intro bind through thiolates, which is a PL-quencher for almost all types of metal-chalcogenide QDs, so mentioning these ligands as a strategy to avoid PL-quenching exciton-delocalizing ligands (in the previous paragraph) makes no sense. Also, there are thiolate exciton-delocalizing ligands (described in papers that the authors did not cite).

Our reply: The ligands such as TOPO, OA, and HDA are often used for the synthesis of oil-soluble QDs by the hot injection method [23],[d,e]. To disperse the QDs in water, it is necessary to exchange these ligands with water-solubilizing ligands. As pointed out by the reviewer, it is known that the thiolates cause PL quenching of QDs during the ligand-exchange process [f,g].

In contrast, for direct synthesis of water-soluble QDs, which does not require ligand-exchange, thiol-based ligands such as TGA and MPA are often used [33-35],[h-j]. It is reported that the QDs directly synthesized in water show high PLQY of more than 60% [33-35],[j]. The PLQY of NAC-capped CdTe QDs used in this study is also very high at 70% [52]. Thus, the concern of PL-quenching can be ruled out in the present NAC-capped CdTe QDs.

References

- [d] Talapin, D. V., Rogach, A. L., Kornowski, A., Haase, M. & Weller, Horst. Highly luminescent monodisperse CdSe and CdSe/ZnS nanocrystals synthesized in a hexadecylamine– trioctylphosphine oxide–trioctylphosphine mixture. *Nano Lett.* **1**, 207-211 (2001).
- [e] Bullen, C. R. & Mulvaney, P. Nucleation and growth kinetics of CdSe nanocrystals in octadecene. *Nano Lett.* **4**, 2303-2307 (2004).
- [f] Munro, A. M., Plante, I. J.-L., Ng, M. S. & Ginger, D. S. Quantitative study of the effects of surface ligand concentration on CdSe nanocrystal photoluminescence. *J. Phys. Chem. C* **111**, 6220-6227 (2007).
- [g] Breus, V. V., Heyes, C. D. & Nienhaus, G. U. Quenching of CdSe–ZnS core–shell quantum dot luminescence by water-soluble thiolated ligands. *J. Phys. Chem. C* **111**, 18589-18594 (2007).
- [h] Gao, M., Kirstein, S., Möhwald, H., Rogach, A. L., Kornowski, A., Eychmüller, A. & Weller H. Strongly photoluminescent CdTe nanocrystals by proper surface modification. *J. Phys. Chem. B* **102**, 8360-8363 (1998).
- [i] N. Gaponik, Talapin, D. V., Rogach, A. L., Hoppe, K., Shevchenko, E. V., Kornowski, A., Eychmüller, A. & Weller H. J Thiol-capping of CdTe nanocrystals: an alternative to organometallic synthetic routes. *Phys. Chem. B* **106**, 7177-7185 (2002).
- [j] Lee, Y.-S., Nakano, K., Bu, H.-B. & Kim, D. Microwave-assisted hydrothermal synthesis of highly luminescent ZnSe-based quantum dots with a quantum yield higher than 90%. *Appl. Phys. Express* **10**, 065001 (2017).

3) I'm not convinced at all by the most important datasets of this paper, the absorption spectra as a function of OD. i think the linewidth of the absorption peak is a better indicator of conjugation among QDs than the peak position. First of all, the peak appears to be broadening with increased monolayer OD, which is not reported here. Second, as the peak broadens it becomes more difficult to pick the peak without a fit (which I also don't see here).

Our reply: To discuss the estimation accuracy of the absorption peak energy pointed out by the reviewer, the absorption spectra of the QD monolayers prepared under two different solution concentrations with $OD_{sol} = 0.02, 0.20$ are shown below as Fig. R1(a). The two spectra were

normalized so that the OD_{mono} values of the absorption peaks coincide. It can be seen that the spectrum of the sample prepared under the condition of $OD_{\text{sol}} = 0.20$, which is shown by solid line, is clearly shifted to the lower energy side than the result of the sample of $OD_{\text{sol}} = 0.02$ shown by dotted line. The absorption peak energies of the sample of $OD_{\text{sol}} = 0.02$ and 0.20 are 2.211 eV and 2.192 eV, respectively. Therefore, the shift amount is estimated to be 19 meV. Figure R1(b) shows the result of shifting the spectrum of $OD_{\text{sol}} = 0.02$ to the low energy side by 19 meV. The entire shape of two spectra are in good agreement. From the results, it can be said that the estimation of the shift amount of the absorption peak is correct.

Next, we discuss the broadening of the absorption peak pointed out by the reviewer. As clearly shown in Fig. R1(b), no broadening of the absorption peak due to the quantum resonance was observed in the CdTe QDSL fabricated by the LBL method. As pointed out by the reviewer, in PbSe QD solids with the ligands exchanged or annealed, the broadening was observed, and the broadening was discussed as the main indicator of QD coupling rather than red shift of absorption peak [16,18],[c]. However, the ligand exchange and/or annealing treatment might induce (1) a change in QD surface, (2) an increase in nonuniformity of QD distance, and (3) a disorder of QD array. All of these factors can cause the broadening of the absorption spectra. Therefore, we think that the broadening does not always become the indicator of the short-range quantum resonance.

In this study, the NAC, which is a short-chain ligand, is used to realize the short-range quantum resonance without the ligand-exchange. A clear absorption peak shift due to the quantum resonance is observed without any peak broadening in our sample. In the previous study [11], we investigated the absorption properties of the QD bilayers in which the QD interlayer distance were different and clarified that the shift amount of the absorption peak energy decreases exponentially with QD interlayer distance. This is a clear evidence that the absorption peak shift is caused by the coupling of wave functions, that is, the short-range quantum resonance. Therefore, in this study, we discuss the shift amount of this absorption peak as an index of the short-range quantum resonance.

We added the above discussion on page 4, lines 15–20 in the revised manuscript.

References

- [c] Jazi, M. A., Janssen, V. A. E. C., Evers, W. H., Tadjine, A., Delerue, C., Siebbeles, L. D. A., H., van der Zant, H. S. J., Houtepen, A. J. & Vanmaekelbergh, D. Transport properties of a two-dimensional PbSe square superstructure in an electrolyte-gated transistor. *Nano Lett.* **17**, 5238-5243 (2017).

Figures R1 | a) The normalized absorption spectra of CdTe QD monolayers prepared under the condition of $OD_{sol} = 0.02$ and $OD_{sol} = 0.20$. b) The spectrum of $OD_{sol} = 0.02$ is shifted to the low energy side by 19 meV.

4) The authors claim that 0.5 nm is short enough to realize mini-band formation. Where does this number come from? Where is the evidence for it? It's completely arbitrary.

Our reply: The quantum resonance arises from the coupling of the wave functions between the adjacent QDs, which is a short-range interaction, and its strength weakens exponentially with the increase in surface-to-surface distance between QDs. In the previous work [11], we observed that the shift amount of the absorption peak energy decreases exponentially with QD interlayer distance (See Fig. 3a in ref. 11). This clearly demonstrates that the observed absorption peak shift is due to the short-range coupling of wave functions, that is, the quantum resonance. When the QD interlayer distance is larger than 2 nm, the absorption peak hardly shifts (See Fig. 3a in ref. 11). Therefore, that the quantum resonance can occur only when the QD distance is approximately 2 nm or less, and the distance of 0.5 nm is sufficiently short to realize miniband formation.

To further prove that the experimental results obtained in this study are due to the formation of miniband by the quantum resonance, we additionally measured the temperature dependence of the PL decay time in the QDSLs in which 1D, 2D, and 3D quantum resonances occur (Figs. S6b, S6c, and S6d). As shown in Figs. S6c and S6d, PL decay times of CdTe QDSLs in which the 1D and 2D quantum resonances occur are proportional to $T^{0.5}$ and $T^{1.0}$, respectively. In ref. 49, Akiyama et al. theoretically and experimentally demonstrated that the PL decay times of quantum wires and quantum wells, in which 1D and 2D electronic states are formed, are proportional to $T^{0.5}$ and $T^{1.0}$, respectively. Therefore, the present temperature dependence of PL decay times clearly demonstrate the formation of the 1D and 2D minibands in the CdTe QDSLs. Furthermore, the PL decay time in the 3D CdTe QDSLs is approximately proportional to $T^{1.5}$, also demonstrating 3D miniband formation.

In this study, by observing no shift in the excitation energy dependence of the PL spectra (Fig. 6) and in the detection energy dependence of the PLE spectra (Fig. 7), and the dimension-dependent temperature dependence of the PL decay time (Fig. S6), as well as the low energy shift of the absorption peak, we demonstrated the formation of miniband derived from the quantum resonance.

We added the above discussion on page 4, lines 15–20 and page 8, lines 18–29 in the revised manuscript, and we added the above discussion on page 2, line 32–page 3, line 3 in the revised supplementary information.

Revised Figure S6

Supplementary Figure S6 | Temperature dependence of radiative lifetime in the QDSL in which 1D, 2D, and 3D quantum resonance occurs. **a**, The dotted lines are PL-decay profiles measured at 80 K (red), 180 K (blue), 250 K (black), respectively. The solid curves are the results of fitting performed using three exponential functions. **b**, The open circles represent the average decay times at each of the temperatures of the QD layer in which 1D quantum resonance occurs; the solid line represents a $T^{0.5}$ dependence. **c**, The open rectangles represent the average decay times at each of the temperatures of the QD layer in which 2D quantum resonance occurs; the solid line represents a $T^{1.0}$ dependence, and dashed line represents a $T^{0.5}$ dependence. **d**, The triangles represent the average decay times at each of the temperatures of the QD layer in which 3D quantum

resonance occurs; the solid line represents a $T^{-1.5}$ dependence, and dashed line represents a $T^{0.5}$ dependence.

5) Similarly, there is not nearly enough evidence or precision in the absorbance spectra to make the quantitative claim that 19 meV is the coupling energy among QDs.

Our reply: As mentioned in the above replies to the reviewer's comments 3 and 4, the estimation of the shift amount of the absorption peak is correct, and it is clear that the low-energy shift of the absorption peak corresponds to the coupling energy among QDs and thus originates from the quantum resonance.

6) The STEM images shown in Figure 3 don't show anywhere near sub-nanometer control of interparticle distance that the authors claim is necessary to tune mini-band formation. The images either show a sub-monolayer or a monolayer. There isn't even enough resolution in the images to determine sub-nanometer distances.

Our reply: In this paper, we do not claim that we can control the distance between QDs with sub-nanometer accuracy. The sub-nanometer control of inter-QD distance has not been achieved. Although it is certain that the average in-plane QD distance changes as the $\sigma_{\text{in-plane}}$ changes, the distance between QDs cannot be controlled yet. However, from the present experimental results about structural analysis and optical properties, we think that there is no doubt that the in-plane quantum resonance occurs in the QD monolayers with high $\sigma_{\text{in-plane}}$. Also, no quantum resonance occurs in the QD monolayers with low $\sigma_{\text{in-plane}}$ because the absorption peak energy coincides with that of the colloidal solution in which the QDs are randomly dispersed.

As described in the manuscript, the average surface-to-surface distance between QDs is obtained by subtracting the QD diameter from the average centre-to-centre distance. Even with the current accuracy, it is quite possible to estimate the average value of the in-plane QD centre distance from the TEM image. Additionally, even with the current resolution in the STEM images shown in Figs. 3a and 3b, it is sufficiently possible to estimate the average centre-to-centre distance. The enlarged image of Fig. 3a was added as an inset (see below), and the centre-to-centre distance between QDs was concretely shown.

We added the following phrases on page 4, line 34 in the revised manuscript: "in the inset in Fig. 3a".

Revised Figure 3a

Figure 3 | Scanning transmission electron microscopy (STEM) images of QD monolayers. a,b, STEM images of QD monolayers with low (a) and high (b) $\sigma_{\text{in-plane}}$, which were fabricated with $\text{OD}_{\text{sol}} = 0.02$ and 0.15 , respectively. **The inset in the figure is the enlarged image. The red dots point to the centre of the QDs, and the red lines connect the points. The numbers shown in white indicate the length of the red lines, which are centre-to-centre distance between QDs. c, d,** Histograms of the centre-to-centre distances between QDs in the QD monolayers with low (c) and high (d) $\sigma_{\text{in-plane}}$.

7) page 5: i don't see any difference in the amount of shift in abs peak between Figures 4a and 4b. The authors need to be clear about how they got their plots in 4c,d. This is subject to a lot of error, without a fit. Also, i would expect a significant baseline for sub-monolayer coverage. Were these spectra collected with an integration sphere? Where is the baseline? Is it subtracted? The subtraction procedure can shift the apparent position of the excitonic peak.

Our reply: To discuss the method of estimating the absorption peak and the method of correcting the absorption spectra pointed out by the reviewer, the discussion of the analysis of absorption spectra and Figs. S2 and S3 were added to SI. The absorption peak energy and the amount of energy shift were estimated by the method described in the reply to the reviewer's comment 3. Figure S2f shows the normalized absorption spectra of CdTe QD monolayer and 3 layers prepared under the condition of $\text{OD}_{\text{sol}} = 0.20$. Figure R2 shows the normalized absorption spectra of CdTe QD monolayers prepared under the conditions of $\text{OD}_{\text{sol}} = 0.02$ and 0.20 , respectively (see below). From both the figures, the low-energy shifts of the absorption peak due to the quantum resonance are clearly observed.

Next, the correction of the absorption spectra will be described. The extinction spectra of CdTe QD monolayer and 3 layers prepared under the condition of $\text{OD}_{\text{sol}} = 0.20$ are shown in Fig. S2a and S2b, respectively. Focusing on the energy region lower than 2.0 eV, it can be seen that the intensity of the extinction spectra is not flat. In other words, as the reviewer pointed out, there is a baseline. Figure S2c shows the extinction spectra of PDDA/PAA multilayers without CdTe QDs and the

extinction spectra of ZnS QD multilayers. Here, ZnS QDs are completely transparent in the photon energy region of 1.75 to 2.75 eV, which is the focus of attention on the absorption properties of CdTe multilayers in this study. Both the normalized extinction spectra are shown in Fig. S2d. It is clear that the two normalized extinction spectra are in good agreement. Furthermore, the extinction spectrum shown in Fig. S2d also coincides with the baseline in the extinction spectra of the CdTe monolayer and 3 layers in Figs. S2a and S2b. These facts mean that this baseline does not reflect the ‘absorption’ of CdTe QDs but reflects the influence of light scattering due to the polymer layer and/or the stacking QDs. Therefore, we obtained the absorption spectra specific to CdTe QD multilayers from the extinction spectra by correcting the baseline caused by scattering. Then, the absorption peak energy was evaluated from the obtained absorption spectra.

To discuss the experimental results for sub-monolayers, the extinction spectra of CdTe QD monolayer prepared under the condition of $OD_{sol} = 0.02$ and 0.20 are shown in Figs. S3a and S3b, respectively. The peak structure is clearly observed even in the extinction spectrum of the sub-monolayer. Furthermore, it can also be seen that the influence of the baseline on the sub-monolayer is small. The normalized extinction spectrum before the baseline correction and the absorption spectrum after the correction are shown in Figs. S3c and S3d. It is evident that the peak energy is almost unchanged before and after the baseline correction.

We added the above discussion on page 3, lines 26–28 and page 10, lines 24–27 in the revised manuscript. In addition, we added the details of the analysis of the absorption spectra on page 1, line 21–page 2, line 10 in the revised supplementary information.

Figure R2 | The normalized absorption spectra of CdTe QD monolayers prepared under the condition of $OD_{sol} = 0.02$ and $OD_{sol} = 0.20$.

Added Figure S2

Supplementary Figure S2 | Analysis of absorption spectra of CdTe QD monolayer and 3 layers with high $\sigma_{in-plane}$. **a,b,** The extinction spectra of CdTe QD (a) monolayer and (b) 3 layers prepared under the condition of $OD_{sol} = 0.20$. The dashed curves show the baseline in the extinction spectra. **c,d,** (c) the extinction spectra of PDDA/PAA multilayers without CdTe QDs and the extinction spectra of ZnS QD multilayers and (d) the normalized spectra. **e,f,** the normalized (e) extinction spectra and (f) absorption spectra, which are corrected, of CdTe QD monolayer and 3 layers prepared under the condition of $OD_{sol} = 0.20$.

Added Figure S3

Supplementary Figure S3 | Analysis of absorption spectra of CdTe QD monolayers with low and high $\sigma_{in-plane}$. **a,b**, The extinction spectra of CdTe QD monolayers prepared under the condition of (a) $OD_{sol} = 0.02$ and (b) $OD_{sol} = 0.20$. The dashed curves show the baselines in the extinction spectra. **c,d**, The dashed and solid curves show the normalized extinction spectra and the absorption spectra, which are corrected, respectively, of the QD monolayers prepared under the condition of (c) $OD_{sol} = 0.02$ and (d) $OD_{sol} = 0.20$. The insets are the enlarged figures in the energy region near the peak.

8) I thought the PLE experiments were supposed to differentiate between the different quantum resonances (1D, 2D, 3D), but they all look the same, so why plot them all?

Our reply: No peak shift was observed in the detection energy dependence of PLE spectra and the excitation energy dependence of PL spectra in all of the three samples with different $\sigma_{in-plane}$ or stacking numbers. It should be noted that regardless of the dimension of the quantum resonance, no shift of PLE or PL peak is observed in the all samples if the coupled states, *i.e.* minibands, are formed. This is why all the results are shown in Figs. 6 and 7.

In this study, we concluded that minibands are formed due to the quantum resonance by systematically investigating the excitation energy dependence of the PL spectra (Fig. 6), the

detection energy dependence of PLE spectra (Fig. 7), and the temperature dependence of the PL decay time (Fig. S6), as well as the low energy shift of the absorption peak. These results are significant in demonstrating that the formation of minibands can be directly observed by experiments other than transport. We believe that this is the clear novelty of our study.

Reply to Reviewer 3

In their manuscript, the authors show the fabrication and optical experiments with superlattices of colloidal CdTe quantum dots. In addition to in-plane 2D assembly, layer-by-layer deposition allows very controlled fabrication of 1D and 3D structures. The results provide evidence that inter-dot electronic coupling establishes a “quantum resonance”; shifted by 19 meV to 39 meV from the isolated QD emission, dependent on the dimensionality. With this degree of control it goes significantly beyond what is there in the literature and is likely to excite a broader audience, especially of course in the colloidal quantum dot community where tailored superlattices have raised increasing attention in recent years.

Overall, the paper is a pleasure to read, and the fabrication and experiments appear very carefully and diligently crafted to yield solid results that are corroborated also by contrasting them to control experiments with isolated QDs. Furthermore, the used methods are well described with sufficient detail, and the samples are thoroughly characterized. Nevertheless, I believe that a small number of points should be addressed before the manuscript is published in Nature Communications.

Our reply: Thank you very much for positively rating our manuscript and suggesting its publication in Nature Communications. We carefully addressed all of your comments and questions. The one-to-one replies are written below. The references listed only in this reply letter are represented by alphabets. In addition, the revised parts of the manuscript are highlighted as red characters in the revised manuscript.

1) Can the authors explain why the energy shift from the quantum resonance going from isolated QDs to 1D is even slightly larger than going from isolated QDs to 2D, i.e. 22 meV vs. 19 meV respectively (Fig. S2). Is the small difference in periodicity (3.9 nm for in-plane and 3.5 nm for out-of-plane structures) sufficient, as the scattering peaks (Fig. 2d and Fig. 5) and also the distributions obtained by STEM (Fig. 3d) are rather broad? Because in principle, I would expect otherwise, i.e. that for the 2D structures the shift should be much larger than the 1D shift.

Our reply: The shift amount of the absorption peak in QDSL in which 1D quantum resonance occurs was slightly larger than that of the QDSL in which 2D quantum resonance occurs. The reason for this is that the in-plane QD distance of 3.9 nm is longer than the interlayer distance of 3.5 nm due to the electrostatic repulsion between the ligands of NACs. In addition, since there is no medium between the in-plane QDs and the medium between the QDs in the stacking direction is PDDA, the presence of PDDA may make the quantum resonance between the QDs in the stacking

direction stronger than in the in-plane direction.

As the reviewer pointed out, the results of the XRD analysis shown in Fig. 2d and Fig. 5 are certainly broad. However, we can see a clear difference in the XRD peak angles when comparing Figures 2d and 5 as shown in Fig. R3 (see below). This clearly shows that the spatial period in the in-plane direction is longer than the stacking direction.

We added the above discussion on page 6, lines 23–28 in the revised manuscript.

Figure R3. Comparison of the out-of-plane XRD pattern of CdTe QD multilayer with high $\sigma_{\text{in-plane}}$ (blue) and the in-plane XRD pattern of CdTe QD monolayer with high $\sigma_{\text{in-plane}}$ (black).

2) While the lifetime vs. temperature dependence of 1D quantum resonance structures show the expected $T^{0.5}$ behavior (Fig. S4) data for the 2D and 3D cases are not shown or mentioned. Ref. 53 suggests e.g. that the 2D case should show T^1 . Such data would substantiate further the claim on the achieved dimensionality.

Our reply: According to the reviewer’s suggestion, we additionally measured the temperature dependence of the PL decay time in the QDSLs in which 1D, 2D, and 3D quantum resonances occur (Figs. S6b, S6c, and S6d, respectively).

As shown in Figs. S6c and S6d, PL decay times in CdTe QDSLs in which the 1D and 2D quantum resonances occur are proportional to $T^{0.5}$ and $T^{1.0}$, respectively. In ref. 49, Akiyama et al. theoretically and experimentally demonstrated that the PL decay times of quantum wires and quantum wells, in which 1D and 2D electronic states are formed, are proportional to $T^{0.5}$ and $T^{1.0}$, respectively. Therefore, the present experimental results of temperature dependence of PL decay time strongly suggest the formation of the 1D and 2D minibands in the CdTe QDSLs. Furthermore, the PL decay time in the 3D CdTe QDSLs is approximately proportional to $T^{1.5}$, demonstrating 3D miniband formation. We believe that these experimental results clearly demonstrate the quantum resonance-based miniband formation in the CdTe QDSLs.

We added the above discussion on page 8, lines 18–29 in the revised manuscript and page 2, line 32–page 3, line 3 in the revised supplementary information.

Revised Figure S6

Supplementary Figure S6 | Temperature dependence of radiative lifetime in the QDSL in which 1D, 2D, and 3D quantum resonance occurs. **a**, The dotted lines are PL-decay profiles measured at 80 K (red), 180 K (blue), 250 K (black), respectively. The solid curves are the results of fitting performed using three exponential functions. **b**, The open circles represent the average decay times at each of the temperatures of the QD layer in which 1D quantum resonance occurs; the solid line represents a $T^{0.5}$ dependence. **c**, The open rectangles represent the average decay times at each of the temperatures of the QD layer in which 2D quantum resonance occurs; the solid line represents a $T^{1.0}$ dependence, and dashed line represents a $T^{0.5}$ dependence. **d**, The triangles represent the average decay times at each of the temperatures of the QD layer in which 3D quantum resonance occurs; the solid line represents a $T^{1.5}$ dependence, and dashed line represents a $T^{0.5}$ dependence.

3) Could a recent model based on altering the dielectric confinement instead of quantum

tunneling (arXiv 1912.04182) provide an alternative explanation for the findings?

Our reply: The dielectric confinement effect pointed out by the reviewer is due to the Coulomb interaction caused by the electric field penetrating into the barrier layer, and the quantum resonance is due to the short-range coupling of wave functions across the barrier layer. Therefore, the former effect is inversely proportional to the centre-to-centre distance between the adjacent nanostructures [a], and the latter effect weakens exponentially with the thickness of the barrier layer [a]. In fact, the low-energy shift amount of the absorption peak calculated by Movilla et al. is inversely proportional to the thickness of the barrier layer [b].

In the previous work [11], we investigated the absorption properties of the QD bilayers in which the QD interlayer distance was controlled in sub-nano scale and observed that the shift amount of the absorption peak energy decreases exponentially with QD interlayer distance (See Fig. 3a in ref. 11). This clearly demonstrates that the observed absorption peak shift is due to the coupling of wave functions, that is, the quantum resonance.

We added the above discussion on page 4, lines 15–20 in the revised manuscript.

References

[a] Kumagai, M. & Takagahara, T. Excitonic and nonlinear-optical properties of dielectric quantum-well structures. *Phys. Rev. B* **40**, 12359 (1989).

[b] Movilla, J. L., Planelles, J. & Climente, J. I. Dielectric confinement enables molecular coupling in stacked colloidal nanoplatelets. *J. Phys. Chem. Lett.* **11**, 3294 (2020). (arXiv 1912.04182)

4) For the low-density in-plane structures that are used for the individual layers in the 1D case, it should be discussed what material actually fills the voids between the QDs. Furthermore, the sketch for the isolated and 1D case in Fig. 1 suggests that there could also be some in-plane periodicity established, while the data in Fig. 3a/c clearly show that this is not the case. It should be possible to improve Fig. 1 to reflect better the lateral randomness.

Our reply: In both QD layers with low $\sigma_{\text{in-plane}}$ and high $\sigma_{\text{in-plane}}$, there is no medium between the in-plane QDs. In the QD layers with low $\sigma_{\text{in-plane}}$, the average in-plane QD distance becomes even longer, but we believe that no substance fills the void.

As pointed out by the reviewer, the picture in the Fig. 1 seemed more regularly arranged than the actual structure. Therefore, we revised Fig. 1.

Revised Figure 1

5) In many of the graphs there are dashed lines to highlight certain features or represent a fit to the data (Fig. 2b, 2d, 4c, 5, 6b-d, 7b-d). It should be briefly described in the respective figure captions.

Our reply: According to the reviewer's comment, we revised the figure captions of Fig. 2b, 2d, 4c, 5, 6b-d, and 7b-d. We appreciate your indication.

REVIEWER COMMENTS

Reviewer #1 (Remarks to the Author):

The authors have addressed my questions to a reasonable extent. They carried out additional temperature-dependent lifetime measurements which exhibit different power laws in different structures. Control experiments were also carried out to demonstrate in long ligand systems, the emission peak position would weakly depend on the excitation wavelength. Overall I appreciate the authors' efforts.

I would recommend the publication of this manuscript if the authors carry out a final set experiments to further verify the effect of dimensionality. The authors should compare QY and lifetime values for (i) solution, (ii) 1D, (iii) 2D, and (iv) 3D structures. In principle, when quantum resonance occurs, excitons can diffuse more freely upon increasing the dimensionality. So I would expect an increase of lifetime and decrease of QY from solution1D2D3D. If it is not the case, the authors should explain.

Reviewer #3 (Remarks to the Author):

In the revised version of the manuscript, the authors significantly enhanced the robustness of the conclusions and the quality of the manuscript. Especially interesting is the new data on the decay times versus temperature that show distinctly different exponents for the different dimensionalities. I am not aware of any alternative scenario like ones based on energy transfer that could explain such peculiar behavior. Furthermore, the authors elucidated their analysis procedure of the absorption data. Importantly, the positioning of the work with respect to previous literature and potential applications has been sharpened, and more details for discerning from alternative interpretations like energy transfer and on the miniband formation have been included in the discussion.

In my view, the criticism has been addressed adequately, making the manuscript now ready to be accepted for publication in Nature Communications.

Minor point:

- In the caption of the revised Supplementary Figure S6 it should be stated that Fig. S6a shows the data for the 1D case.

Reply to Reviewer 1

The authors have addressed my questions to a reasonable extent. They carried out additional temperature-dependent lifetime measurements which exhibit different power laws in different structures. Control experiments were also carried out to demonstrate in long ligand systems, the emission peak position would weakly depend on the excitation wavelength. Overall I appreciate the authors' efforts.

I would recommend the publication of this manuscript if the authors carry out a final set of experiments to further verify the effect of dimensionality. The authors should compare QY and lifetime values for (i) solution, (ii) 1D, (iii) 2D, and (iv) 3D structures. In principle, when quantum resonance occurs, excitons can diffuse more freely upon increasing the dimensionality. So I would expect an increase of lifetime and decrease of QY from solution1D2D3D. If it is not the case, the authors should explain.

Our reply: Thank you very much for positively rating our revised manuscript and suggesting its publication in Nature Communications.

Additional experiments were performed following the reviewer's advice. The QYs for solution samples and 1D, 2D, and 3D structures are 78, 27, 26, and 22 %, respectively, and the lifetimes are 30, 2.7, 3.6, and 4.4 ns, respectively. There is a big jump in the QYs and lifetimes between the solution samples and the QDSLs. In the QDSLs, where the minibands are formed, an additional relaxation process such as intra-band relaxation peculiar to the QDSLs would appear. Therefore, the shorter lifetime of the QDSLs compared to that of the solution sample seems a reasonable result. In addition, the PL QY in the QDSLs is lower than that in the solution sample due to the free diffusion of excitons in the QDSLs (as the referee pointed out) and the existence of an additional relaxation process peculiar to the QDSLs.

When compared among the QDSLs, the PL QY decreases slightly, and the lifetime increases slightly as the dimensionality increases. This seems to reflect the referee's point that "*excitons can diffuse more freely upon increasing the dimensionality*". We are very grateful to you for making us aware of this important point.

We added the above discussion on page 8, lines 24-29 in the revised manuscript. Also, we added the description about experimental procedures of PL QY and PL decay profiles to *Method*.

Added: page 8, lines 24-29

The PL QYs for 1D, 2D, and 3D structures are 27, 26, and 22 %, respectively, and the PL decay times are 2.7, 3.6, and 4.4 ns, respectively. The PL QY decreases slightly, and the PL decay time increases slightly as the dimensionality increases. These results suggest that excitons can be more widely delocalised owing to the stronger quantum resonance with the higher dimensionality. It is

expected that the radiation mechanism in QDSLs will be further clarified by examining the transient absorption and its temperature dependence in detail.

Reply to Reviewer 3

In the revised version of the manuscript, the authors significantly enhanced the robustness of the conclusions and the quality of the manuscript. Especially interesting is the new data on the decay times versus temperature that show distinctly different exponents for the different dimensionalities. I am not aware of any alternative scenario like ones based on energy transfer that could explain such peculiar behavior. Furthermore, the authors elucidated their analysis procedure of the absorption data. Importantly, the positioning of the work with respect to previous literature and potential applications has been sharpened, and more details for discerning from alternative interpretations like energy transfer and on the miniband formation have been included in the discussion.

In my view, the criticism has been addressed adequately, making the manuscript now ready to be accepted for publication in Nature Communications.

Minor point:

- In the caption of the revised Supplementary Figure S6 it should be stated that Fig. S6a shows the data for the 1D case.

Our reply: Thank you very much for positively rating our revised manuscript and suggesting its publication in Nature Communications.

We carefully checked the caption of Supplementary Figure S6 and revised it. We appreciate your indication.

Reviewers' comments:

Reviewer #1 (Remarks to the Author)

The authors have addressed my questions to a reasonable extent. They carried out additional temperature-dependent lifetime measurements which exhibit different power laws in different structures. Control experiments were also carried out to demonstrate in long ligand systems, the emission peak position would weakly depend on the excitation wavelength. Overall I appreciate the authors' efforts.

I would recommend the publication of this manuscript if the authors carry out a final set experiments to further verify the effect of dimensionality. The authors should compare QY and lifetime values for (i) solution, (ii) 1D, (iii) 2D, and (iv) 3D structures. In principle, when quantum resonance occurs, excitons can diffuse more freely upon increasing the dimensionality. So I would expect an increase of lifetime and decrease of QY from solution1D2D3D. If it is not the case, the authors should explain.

Reviewer #3 (Remarks to the Author)

In the revised version of the manuscript, the authors significantly enhanced the robustness of the conclusions and the quality of the manuscript. Especially interesting is the new data on the decay times versus temperature that show distinctly different exponents for the different dimensionalities. I am not aware of any alternative scenario like ones based on energy transfer that could explain such peculiar behavior. Furthermore, the authors elucidated their analysis procedure of the absorption data. Importantly, the positioning of the work with respect to previous literature and potential applications has been sharpened, and more details for discerning from alternative interpretations like energy transfer and on the miniband formation have been included in the discussion.

In my view, the criticism has been addressed adequately, making the manuscript now ready to be accepted for publication in Nature Communications.

Minor point:

- In the caption of the revised Supplementary Figure S6 it should be stated that Fig. S6a shows the data for the 1D case.

Reply to Reviewer 1

The authors nicely addressed my last question. I am happy to support publication of this manuscript.

Our reply: Thank you very much for positively rating our revised manuscript and suggesting its publication in Nature Communications. We appreciate a lot of your constructive advice to improve the quality of the article.